# Comprehensive analysis of coding variants highlights genetic complexity in developmental and epileptic encephalopathy

Atsushi Takata et al.[#]

Although there are many known Mendelian genes linked to epileptic or developmental and epileptic encephalopathy (EE/DEE), its genetic architecture is not fully explained. Here, we address this incompleteness by analyzing exomes of 743 EE/DEE cases and 2366 controls. We observe that damaging ultra-rare variants (dURVs) unique to an individual are significantly overrepresented in EE/DEE, both in known EE/DEE genes and the other non-EE/DEE genes. Importantly, enrichment of dURVs in non-EE/DEE genes is significant, even in the subset of cases with diagnostic dURVs ($P = 0.000215$), suggesting oligogenic contribution of non-EE/DEE gene dURVs. Gene-based analysis identifies exome-wide significant ($P = 2.04 \times 10^{-6}$) enrichment of damaging de novo mutations in *NF1*, a gene primarily linked to neurofibromatosis, in infantile spasm. Together with accumulating evidence for roles of oligogenic or modifier variants in severe neurodevelopmental disorders, our results highlight genetic complexity in EE/DEE, and indicate that EE/DEE is not an aggregate of simple Mendelian disorders.

[#]A full list of authors and their affiliations appears at the end of the paper.

Epileptic encephalopathy (EE) is a severe disabling condition characterized by specific electroencephalographic patterns and developmental delay/regression that is assumed to be contributed by epileptiform abnormalities themselves[1]. In practice, often it is difficult to clarify whether the epileptic or developmental component more significantly contributes to the clinical manifestations. The term developmental and epileptic encephalopathy (DEE) was thereby introduced recently to refer to conditions where both epileptic and developmental mechanisms are supposed to be involved in the pathogenesis[2]. Regarding the etiology of EE/DEE, it has been well established that genetic factors play important roles. To date, dozens of genes convincingly linked to DEE have been identified[3]. On the other hand, even after performing comprehensive screening of gene-disruptive single nucleotide and copy number variations (i.e. whole exome sequencing (WES) and microarray analyses), typically obtained diagnostic yields are in the range of 25–40%[4,5]. In addition, often there is substantial phenotypic variability among carriers of mutations in the same gene, or even among carriers of the same mutation[6,7], indicating existence of unidentified factors explaining this variation. Such limitations in the understanding of the genetic architecture of EE/DEE can be addressed by conducting large-scale comprehensive analyses. Indeed, WES studies focusing on de novo mutations (DNMs) have provided plenty of information on previously unrecognized disease genes as well as biological pathways implicated in EE/DEE beyond ion channels[8,9]. Besides these analyses focusing on DNMs, another major approach in statistical genetics, namely the case−control analysis, should provide additional insights into the genetic landscape of EE/DEE.

Here we analyze WES data of 3109 individuals of Japanese origin (743 EE/DEE cases and 2366 controls). We investigate both rare and common exonic variants with an emphasis on ultra-rare variants (URVs: defined as variants only once seen in our cohort and not found in databases of variants in the general population). We observe significant excess of damaging URVs in EE/DEE, both in known EE/DEE genes and the other non-EE/DEE genes. In addition, we found unexpected enrichment of damaging URVs in non-EE/DEE genes even in the subset of EE/DEE cases with diagnostic variants in known EE/DEE genes. This result suggests that these non-EE/DEE gene damaging URVs could contribute to the full manifestation of EE/DEE in an oligogenic manner. Overall, our results provide multiple lines of experimental evidence indicating that EE/DEE is not an aggregate of simple Mendelian disorders, highlighting the genetic complexity in EE/DEE.

## Results

**Patterns of excess of URVs in EE/DEE.** In this study, we analyzed an extensively quality-controlled (Supplementary Figs. 1−3) dataset of 3109 exomes (743 EE/DEE cases and 2366 controls, detailed information of EE/DEE subtypes is available in Supplementary Table 1) of Japanese origin. By extracting variants that were only once observed in our overall case−control cohort and never seen in the Exome Aggregation Consortium (ExAC)[10], the Exome Sequencing Project (ESP)[11], nor the Tohoku Medical Megabank Organization (ToMMo)[12] databases, we identified a total of 169,014 and 42,974 URVs in coding and noncoding regions, respectively. While we were aware that recurrent pathogenic mutations and true disease-contributing variants observed in general populations at a low frequency would be removed by restricting the primary scope of our analysis to these URVs[13], this procedure enables us to do an unbiased analysis that can efficiently detect enrichment of rare damaging variants in other neurodevelopmental disorders[14−16].

To understand general characteristics of various functional types of URVs observed in EE/DEE, we stratified these variants according to their functionality into null (nonsense, frameshift, splice site and read-through, also referred to as loss-of-function (LOF)), Moderate (defined by SnpEff[17]; e.g. missense and inframe indel; all of these variants were included in this type regardless of their predicted deleteriousness by in silico tools at this stage of classification), synonymous and noncoding variants (see Supplementary Table 2, Methods and SnpEff manual page (Effect Prediction Details section of http://snpeff.sourceforge.net/SnpEff_manual.html) for details). The numbers of null, Moderate, synonymous, and noncoding URVs were 12,032, 109,063, 47,919, and 42,974, respectively. We then subjected them to logistic regression analysis testing for association of per-individual numbers of each type of URVs and the case−control status. We found significant excess of null and Moderate URVs in EE/DEE (Fig. 1a, null, $P = 0.0000168$, odds ratio (OR) = 1.09; Moderate, $P = 0.00414$, OR = 1.02, note that $P$ values described in this manuscript are raw, uncorrected $P$ values if not specified), nominal ($P > 0.05$ when Bonferroni-corrected with the number of URV types tested in each panel of a figure) excess of synonymous URVs in the cases ($P = 0.0367$, OR = 1.02), and no significant association of noncoding URVs ($P = 0.266$, OR = 1.01). For Moderate URVs, recent studies have demonstrated that variants predicted to be damaging by multiple in silico prediction algorithms (Methods, we refer to them as consensus-damaging (CD) missense URVs) are particularly enriched in patients with neurodevelopmental disorders[14,15,18]. According to these observations, we stratified Moderate URVs into CD missense ($n = 12,337$) and the other missense/inframe URVs ($n = 96,726$). We observed highly significant excess of CD missense URVs in EE/DEE (Fig. 1a, $P = 0.0000643$, OR = 1.09), whereas there was no significant excess of the other missense/inframe URVs ($P = 0.0861$, OR = 1.01). Regarding synonymous URVs, we observed nominally significant excess in EE/DEE (Fig. 1a). To further inspect possible source of this result, we stratified synonymous URVs into those within DNase I hypersensitive sites in frontal cortex (FCDHS synonymous, $n = 11,247$), which were demonstrated to be potentially associated with neurodevelopmental disorders[19], splice region synonymous URVs (SR synonymous, $n = 1277$) defined as those at the last and the first 3 bp of an exon adjacent to an intron, and the other synonymous variants less likely to be functional ($n = 35,608$). We found nominally significant excess of FCDHS synonymous URVs in EE/DEE (Fig. 1a, $P = 0.0251$, OR = 1.05). While SR synonymous URVs are not significantly enriched in EE/DEE, the observed OR was similar to that for FCDHS synonymous URVs ($P = 0.388$, OR = 1.06). For the other synonymous URVs, there was no significant excess with an OR close to one ($P = 0.278$, OR = 1.01). According to these patterns observed for subsets of synonymous URVs, nominally significant enrichment in the synonymous group would be explained by potentially functional (i.e. FCDHS and SR synonymous) URVs rather than artifacts such as population stratification.

We next analyzed patterns of excess of URVs in specific gene sets. When we focused on genes intolerant to null variants in the general population of ExAC[10] (probability of being LOF-intolerant (pLI) > 0.9, $n$ of genes = 3230), we observed significant excess of likely functional URVs (e.g. null, CD missense and FCDHS synonymous, Fig. 1b), with ORs that were consistently higher than what we have observed in the analysis of URVs in all genes (Fig. 1b). Expectably, further prominent excess of likely functional URVs in EE/DEE was observed in established autosomal dominant or X-linked EE/DEE genes (Supplementary Table 3; $n$ of genes = 58, compiled by our manual literature search and from refs. [9,20]; we refer to them as 58EE/DEE genes in

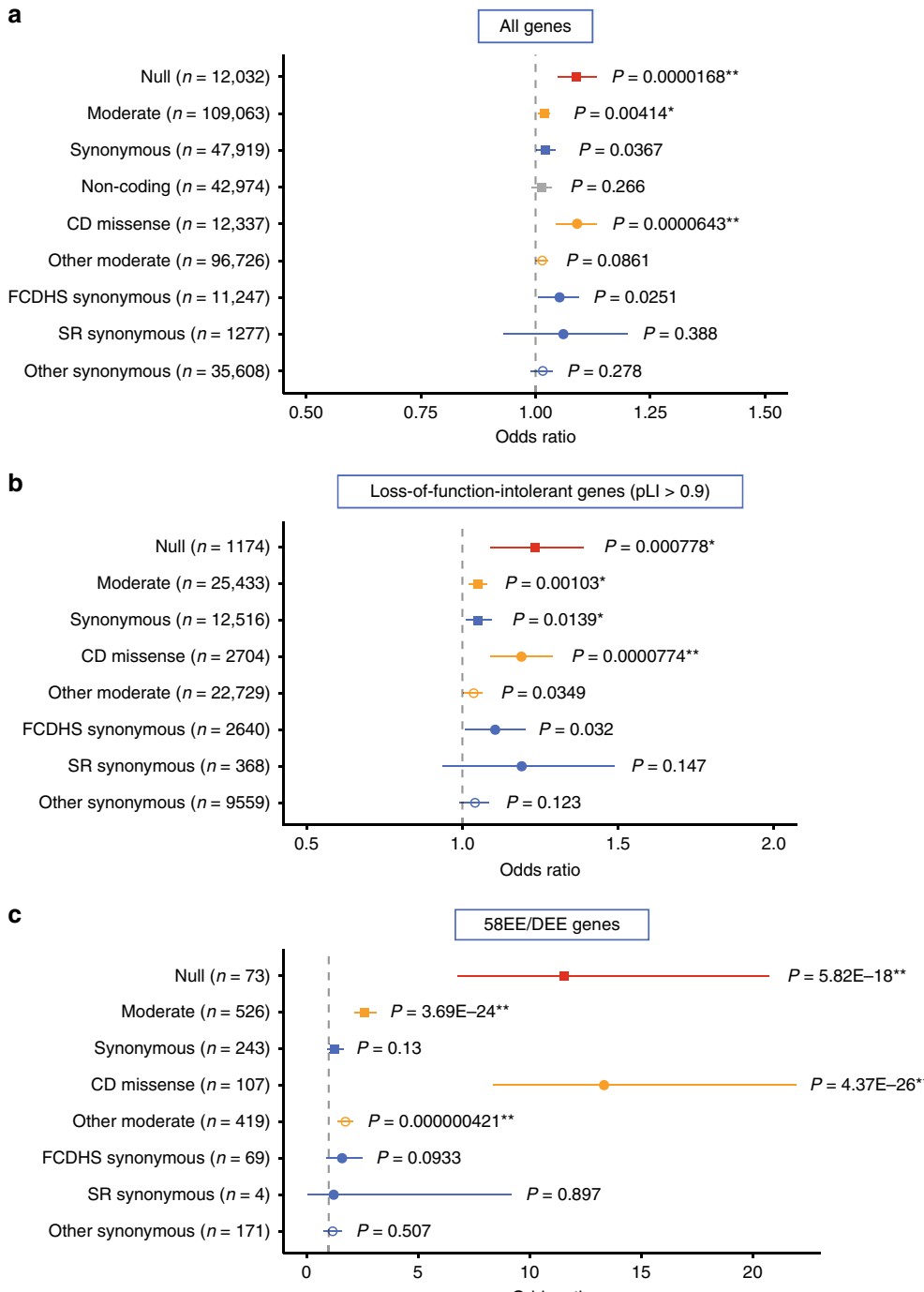

**Fig. 1** Patterns of excess of URVs in EE/DEE. **a** Result of logistic regression analysis testing association between each type of URVs and the case–control status. Odds ratios for one additional URV and 95% confidence intervals are plotted (i.e. one additional URV changes the risk of being EE/DEE with the indicated odds ratio). Uncorrected P values for each test are shown beside the plots. Plots are color-coded as follows: null, red; Moderate (defined by SnpEff[17], e.g. missense and inflame), orange; synonymous, blue; noncoding, gray; and shape-coded as follows: overall functional type with no subclassification (e.g. null and Moderate), filled square; URVs more likely to be functional (i.e. consensus-damaging (CD) missense and frontal cortex DNase I hypersensitive site (FCDHS) and splice region (SR) synonymous), filled circle; URVs less likely to be functional, open circle. The numbers of URVs subjected to each analysis are indicated in the brackets. **b**, **c** Results of logistic regression analysis focusing on LOF-intolerant genes (probability of being LOF-intolerant (pLI) > 0.9) (**b**) and known 58EE/DEE genes (Supplementary Table 3) (**c**), respectively. Statistically significant results considering the total number of hypotheses tested in Figs. 1 and 2 (n = 66) are indicated as follows: **Bonferroni-corrected P < 0.05 (raw P < 0.000758), *Benjamini–Hochberg-corrected P < 0.05 (raw P < ~0.02)

the following) (Fig. 1c). Another important thing is that null or CD missense URVs in 58EE/DEE genes, which would be considered as pathogenic when identified in an EE/DEE case, were not absent in the controls. We indeed identified 39 such URVs in controls (Supplementary Data 1), indicating

need of careful interpretation in clinical settings. Also this observation may provide some insights into resilience against pathogenic variants. More detailed properties of these URVs in 58EE/DEE genes found in controls are discussed in Supplementary Note 1.

**Analyses of EE/DEE cases with or without pathogenic URVs.** As described above, there was striking excess of null and CD missense URVs in 58EE/DEE genes in cases (based on this observation, we refer to the combined group of null and CD missense URVs as damaging URVs (dURVs) in the following). We next analyzed the set of the other non-58EE/DEE genes. In our overall cohort, 18,396 protein-coding genes (based on ENSG IDs annotated by SnpEff[17]) were with one or more URVs, and thereby there were 18,338 (18,396-58) analyzable non-58EE/DEE genes. We found that dURVs in non-58EE/DEE genes are still significantly overrepresented in EE/DEE group (Fig. 2a, null, $P = 0.000392$, OR = 1.07; CD missense, $P = 0.00543$, OR = 1.06; dURV, $P = 9.5 \times 10^{-6}$, OR = 1.07), indicating that analysis of dURVs in non-58EE/DEE genes can aid discovery of new EE/DEE genes and/or provide additional evidence to genes whose association with EE/DEE has not been established. In this context, we stratified the case cohort into individuals carrying dURVs in 58EE/DEE genes and the others, assuming that by focusing on individuals without such dURVs in known genes we would be able to increase the chance to identify new EE/DEE genes. Among individuals without dURVs in 58EE/DEE genes ($n$ of individuals = 605), we observed expectable enrichment of dURVs in non-58EE/DEE genes (Fig. 2b, null, $P = 0.00285$, OR = 1.07; CD missense, $P = 0.0174$, OR = 1.06; dURV, $P = 0.000162$, OR = 1.06). We next analyzed cases with dURVs in 58EE/DEE genes ($n = 138$). Unexpectedly, we found that in this subpopulation there was significant excess of dURVs in non-58EE/DEE genes (Fig. 2c, null, $P = 0.0130$, OR = 1.11; CD missense, $P = 0.0612$, OR = 1.09; dURV, $P = 0.00227$, OR = 1.09). This was true when we further extracted individuals carrying convincingly pathogenic URVs (pURVs), that is, dURVs in 58EE/DEE genes confirmed to be de novo or reasonably transmitted from one of the unaffected parents in the instances of X-linked genes ($n = 116$, Supplementary Data 2, see Methods for more details) (Fig. 2d, null, $P = 0.00592$, OR = 1.13; CD missense, $P = 0.0107$, OR = 1.13; dURV, $P = 0.000215$, OR = 1.12). These observations were unlikely to be fully explained by global genomic instability in these individuals, younger ages in cases when compared with controls, and/or other unknown/unidentifiable confounding factors (Supplementary Note 2).

Though we understand that further stratification reduces statistical power, we repeated the analysis by separating the EE/DEE cases with pURVs in 58EE/DEE genes ($n = 116$) into carriers of null pURVs ($n = 45$) and CD missense pURVs ($n = 71$). We observed that there was no statistically significant excess of dURVs in non-58EE/DEE genes among the carriers of null pURVs (Fig. 2e, $P = 0.138$, OR = 1.08) that are expected to uniformly disrupt gene function (and may not require additional factors for full clinical manifestation). By contrast, dURVs in non-58EE/DEE genes were significantly enriched among the carriers of CD missense pURVs in 58EE/DEE genes ($P = 0.000321$, OR = 1.15), whose impact on gene/protein function would be more variable when compared with null alleles. Among the 116 pURVs in 58EE/DEE genes, we found that 49 are registered in the Human Gene Mutation Database (HGMD, version2017.3) (Supplementary Data 2). The other pURVs were not in HGMD, and can be used as a resource of previously unidentified likely pathogenic mutations. Of the pURVs registered in HGMD, 33 were observed in patients with EE/DEE, whereas ten pURVs were observed in non-EE/DEE phenotypes such as autism with no description of epilepsy. When we stratified 116 individuals with pURVs into carriers of pURVs previously observed in non-EE/DEE ($n = 10$), observed in EE/DEE ($n = 33$) and the others ($n = 73$) (Supplementary Table 4 and Methods), we found enrichment of dURVs in non-58EE/DEE genes with a high OR among the carriers of pURVs that were previously observed in non-EE/DEE phenotypes (Fig. 2f, $P = 0.0108$, OR = 1.29). Because pURVs previously reported in non-EE/DEE may not fully explain the EE/DEE phenotypes observed in our cohort, additional dURVs in non-58EE/DEE genes would be contributing to the full manifestation of EE/DEE in an oligogenic manner. Meanwhile, enrichment of dURVs in non-58EE/DEE genes was nonsignificant among the carriers of pURVs previously observed in EE/DEE ($P = 0.184$, OR = 1.08), and was significant with an intermediate OR among the carriers of the other pURVs ($P = 0.00376$, OR = 1.12). A schema of the procedures for stratification of the case group is shown in Fig. 2g.

After performing these analyses, we evaluated which of the above findings are statistically robust by applying a multiple testing correction considering the total number of hypotheses evaluated in Figs. 1 and 2 ($n$ of hypotheses = 66). When we applied a stringent Bonferroni correction, 12 types of URVs remained significant (Supplementary Data 3, the significance threshold for raw $P$ value = 0.000758). Specifically, enrichment of dURVs in non-58EE/DEE genes among the 116 individuals with pURVs in 58EE/DEE genes was significant after correction (uncorrected $P = 0.000215$, Bonferroni-corrected $P = 0.0142$). We then performed a correction with Benjamini−Hochberg procedure. This correction would be a more balanced approach, considering that many of the 66 tests in our study are not independent of others, and we analyzed several negative control variant types (e.g. likely nonfunctional synonymous variants). We found that 26 types of URVs show significant enrichment after correction (Supplementary Data 3, the significance threshold for raw $P$ value = ~0.02). The Benjamini−Hochberg-corrected $P$ value for enrichment of dURVs in non-58EE/DEE genes among the pURVs carriers was 0.00142.

To explore the properties of dURVs in non-58EE/DEE genes among the 116 pURV carriers (i.e. potential modifier/oligogenic URVs), we performed an analysis incorporating gene expression data of various tissue/cell types in the Genotype-Tissue Expression (GTEx) project[21]. For this analysis, we first constructed lists of genes specifically expressed in each tissue, and then tested association of dURV counts in genes specific to each tissue and the case−control status using the data of dURVs in non-58EE/DEE genes among the 116 pURV carriers and 2366 controls (Methods). We found that six out of the top ten tissues were brain regions (Supplementary Fig. 4a, Brain-Cerebellum, Brain-Cerebellar Hemisphere, Brain-Cortex, Brain-Frontal Cortex (Brodmann Area 9), Brain-Hippocampus and Brain-Anterior cingulate cortex (Brodmann Area 9)). Brain regions were with significantly smaller logistic regression $P$ values for enrichment of non-58EE/DEE gene dURVs among the 116 pURV carriers, when compared with the other tissues ($P = 0.0386$, two-tailed Wilcoxon rank sum test). This observed pattern across various tissues was largely replicated when lists of genes with moderate to high expression in each tissue (transcripts per million reads (TPM) > 10 in the GTEx data) was used (Supplementary Fig. 4b, $P = 0.00334$, two-tailed Wilcoxon rank sum test comparing brain regions and the other tissues). These results further support roles of non-58EE/DEE gene dURVs among the 116 pURV carriers in brain function/development. The list of dURVs in non-58EE/DEE genes among the 116 pURV carriers is available in Supplementary Data 4. We note that there was one individual who carries two dURVs in the 58EE/DEE genes, a de novo CD missense variant in *GRIN1* (c.2506G > C [p.Gly836Arg]) and a maternally inherited CD missense variant in *SIK1* (c.571G > A [p.Ala191Thr]).

**Analyses of doubleton/tripleton and common variants.** We consequently analyzed doubleton and tripleton rare variants, that is, variants observed twice and three times in our overall case

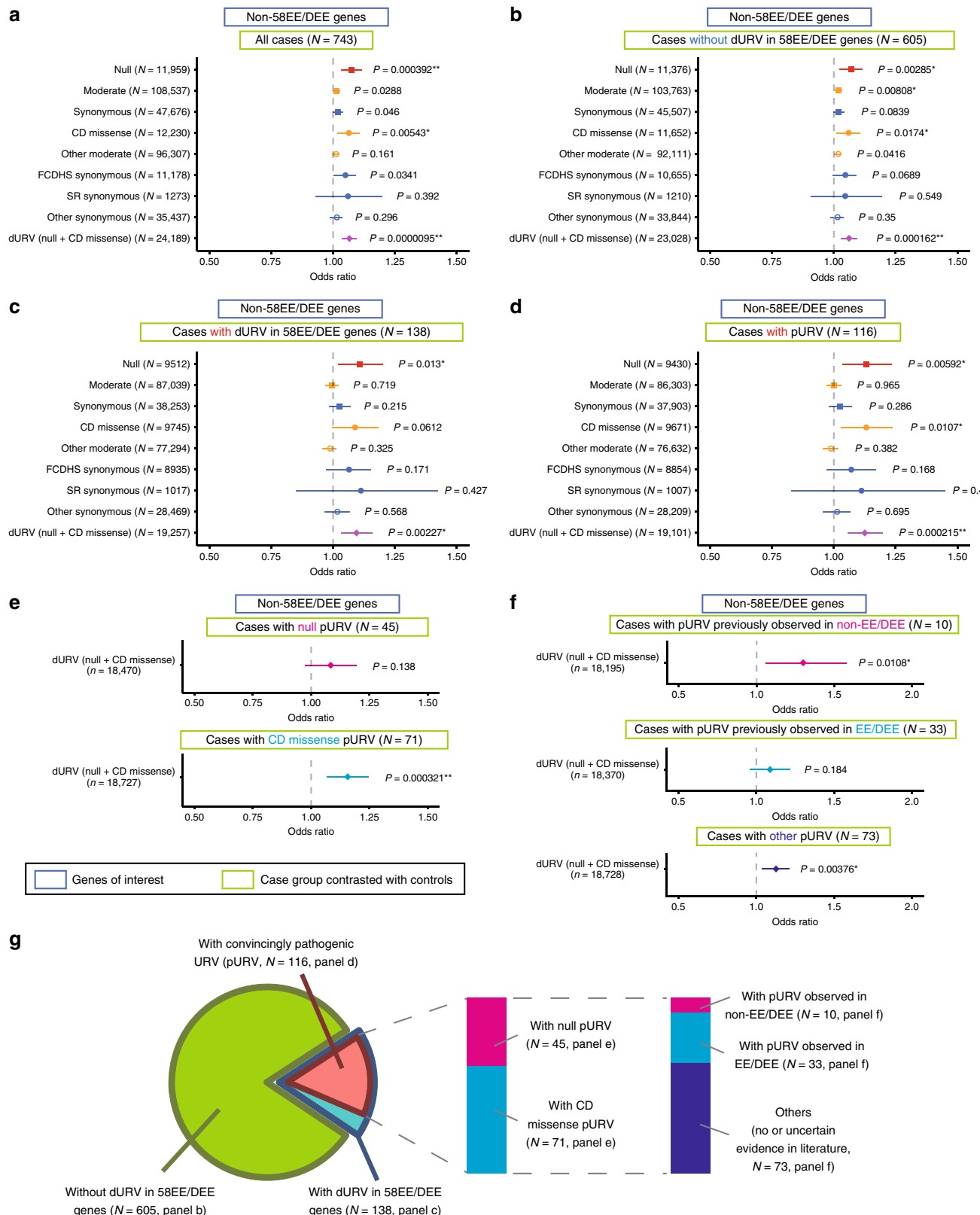

−control cohort, respectively, and not seen in any of ExAC, ESP, and ToMMo databases. As a whole, we did not observe overall enrichment of any functional types of doubleton/tripleton variants in EE/DEE (Supplementary Fig. 5). The total number of convincingly pathogenic doubleton/tripleton alleles ($n = 24$, Supplementary Table 5) was smaller than singleton pURVs ($n =$

116). More detailed results of analyses of doubleton/tripleton rare variants are described in Supplementary Note 3.

When we performed an exome-wide association study of single nucleotide polymorphisms (SNPs: defined as variants with a minor allele frequency (MAF) >1%), we found that there is no exonic SNP surpassing the genome-wide significance threshold

**Fig. 2** Excess of dURVs in non-58EE/DEE genes in individuals with or without pathogenic mutations. **a** Result of logistic regression analysis testing association between each type of URVs in genes not included in 58EE/DEE genes (non-58EE/DEE genes) and the case−control status. Odds ratios for one additional URV and 95% confidence intervals are plotted. Uncorrected *P* values for each test are shown beside the plots. Plots are color-coded as shown in the legend of Fig. 1. The combined group of damaging (null and CD missense) URVs (dURVs) was indicated by the purple diamond. **b−d** Results of analyses of URVs in non-58EE/DEE genes comparing control subjects with the following subsets of case group: cases not carrying dURVs in 58EE/DEE genes (**b**, *n* = 605), cases carrying dURVs in 58EE/DEE genes (**c**, *n* = 138), and cases carrying convincingly pathogenic URVs (pURVs; e.g. confirmed de novo mutations in 58EE/DEE genes; see the main text and Methods) (**d**, *n* = 116). **e**, **f** Results of analyses stratifying cases with pURVs in 58EE/DEE genes into subsets. In (**e**), groups of null pURV carriers (*n* = 45) and CD missense pURV carriers (*n* = 71) were analyzed for the burden of dURVs in non-58EE/DEE genes. In (**f**), we analyzed groups of individuals carrying; pURVs previously reported in non-EE/DEE phenotypes (*n* = 10, based on information in the Human Gene Mutation Database), pURVs previously reported in EE/DEE (*n* = 33), and the other pURVs with no or uncertain previous report (*n* = 73). In (**a−f**), genes of interest included in the analysis are indicated in the blue boxes (as in Fig. 1), and the case subsets contrasted with the controls are indicated in the light green boxes. The number of URVs analyzed in each test is indicated in the brackets. Statistically significant results considering the total number of hypotheses tested in Figs. 1 and 2 (*n* = 66) are indicated as follows: **Bonferroni-corrected *P* < 0.05 (raw *P* < 0.000758), *Benjamini−Hochberg-corrected *P* < 0.05 (raw *P* < ~0.02). **g** A diagram indicating relationship among the case subsets subjected to the analyses

---

($P < 5 \times 10^{-8}$) (Supplementary Fig. 6 and Supplementary Data 5). More detailed results of the exome-wide association study, including calculation of statistical power (Supplementary Fig. 7), are shown in Supplementary Note 4.

**Gene-based burden test and comparison with common epilepsies.** As indicated by the overall enrichment of dURVs in non-58EE/DEE genes, gene-based analysis of URVs can aid discovery of new EE/DEE genes. We therefore performed a gene-based burden test of dURVs using a collapsing method[22]. We did not include doubleton and tripleton rare variants as there was no significant overall enrichment of null or CD missense doubleton/tripleton rare variants in non-EE/DEE genes (Supplementary Fig. 5g, h). We first included all EE/DEE individuals and observed exome-wide significant ($P < 0.05/20,000 = 2.5 \times 10^{-6}$) burden of dURVs in *CDKL5*, *STXBP1*, *SCN1A*, *SCN2A* and *KCNQ2* (Fig. 3a and Supplementary Data 6); all of these are well-established EE/DEE genes. Genes with nominal significance ($2.5 \times 10^{-6} \leq P < 0.05$, *n* of genes = 97) were also highly enriched for 58EE/DEE genes (Fig. 3b, $P < 0.0001$, calculated by random shuffling of the case−control status, see Methods for details), suggesting existence of previously unrecognized EE/DEE genes among the other nominally significant genes. We then repeated the analysis excluding EE/DEE cases with pURVs (*n* of individuals = 116, see above for the definition of pURV), because majority of these variants should have a primary pathogenic effect. At the single-gene level, there was no gene reached to the exome-wide significance ($P = 2.5 \times 10^{-6}$). Most significant enrichment was observed for *NF1* ($P = 0.000477$, OR = 22.8), the gene causal for neurofibromatosis type 1 (NF1), followed by *TRPM5*, *AP5B1*, *DNMT3L* and *ARFGEF1* (Supplementary Data 6). By subjecting nominally significant genes ($P < 0.05$, n = 193, not including 58EE/DEE genes) to a gene ontology (GO) enrichment analysis using ToppGene[23], we identify a total of 27 GO terms enriched among the input genes (Benjamini−Hochberg-corrected $P < 0.05$, Supplementary Table 6). The most significantly enriched term was GO:0006811:ion transport (Benjamini−Hochberg-corrected $P = 0.00266$), which includes genes very recently implicated in neurodevelopmental phenotypes, such as *CAMK2A* and *PPP3CA*[24–27]. When networks of significantly enriched GO terms were delineated, we observed a large cluster of terms related to transportation of ions such as calcium and a small cluster of terms related to homeostatic processes (Fig. 3c). By comparing the results of gene-based burden tests for EE/DEE in our study and a recent study for common forms of epilepsy (genetic generalized epilepsy: GGE and nonacquired focal epilepsy: NAFE)[20], we found *SCN1A*, *KCNQ2*, *ATP1A3* and *GRIA4* as genes with nominally significant burden of rare damaging variants in both EE/DEE and common epilepsy (Supplementary Fig. 8a and

Supplementary Data 7). We also found significant gene-level overlap of EE/DEE with GEE (Supplementary Fig. 8b, $P = 0.0115$), but not with NAFE (Supplementary Fig. 8b, $P = 0.373$) (see Supplementary Note 5 for more details).

**Enrichment of damaging *NF1* DNMs in infantile spasm.** In the gene-based burden test, we identified *NF1* as the gene with the smallest *P* value among non-58EE/DEE genes. Because DNA samples of the parents of EE/DEE patients with *NF1* dURVs were available, we tested if these URVs are DNMs or inherited from one of the parents. We confirmed that three dURVs (Fig. 3d and Supplementary Data 8, c.3445A > G [p.Met1149Val], c.4835 + 1G > T and c.5330T > A [p.Val1777Asp]) are DNMs. Based on an established model of per-gene DNM rates[28], we calculated that the probability of observing three or more damaging DNMs in *NF1* in 627 probands as $4.66 \times 10^{-6}$. *P* values corrected for multiple testing (Bonferroni procedure) with the number of testable genes in our study (*n* = 10,800, genes with one or more dURVs) and ~20,000 all protein-coding genes were 0.0503 and 0.0932, respectively. In addition, we note that all probands carrying a damaging DNM in *NF1* were diagnosed with infantile spasm. The probability of observing three or more damaging DNMs in the subset of infantile spasm probands in our cohort (*n* = 237, pURV carriers were excluded) was $2.55 \times 10^{-7}$, which surpasses the exome-wide significance threshold after correcting for the number of EE/DEE subtypes analyzed (corrected $P = 2.04 \times 10^{-6}$, *n* of EE/DEE subtypes = 8, Supplementary Table 1). Brief overview of the clinical manifestations in these three individuals with a damaging *NF1* DNM is shown in Supplementary Table 7.

**Analysis of other selected candidate genes.** Besides *NF1*, we tested inheritance patterns of dURVs in four other genes, *ARFGEF1*, *STXBP5L*, *HUWE1*, and *CACNA1E*; all of which showed nominally significant excess of dURVs in EE/DEE in our analysis and are highly intolerant against null variants in the general population (pLI score > 0.99) (Supplementary Data 8). While we did not observe exome-wide significant enrichment of damaging DNMs in these genes, partly due to limited availability of the DNA samples of the parents, we confirmed that two CD missense URVs in *CACNA1E*, a gene recently reported as a EE/DEE/neurodevelopmental disorder gene[9,29], are DNMs. Of the confirmed two DNMs in this study, the c.2104G > A [p.Ala702Thr] variant was reported as recurrent mutations in the above-mentioned study[29], and the c.2092T > C [p.Phe698Leu] variant was a mutation at an amino acid residue where another substitution (p.Phe698Ser) was reported[29]. In accordance with the recently reported cases with *CACNA1E* mutations[29], we observed congenital contractures and dystonia in both of our cases (brief summary of the clinical phenotypes is available in Supplementary

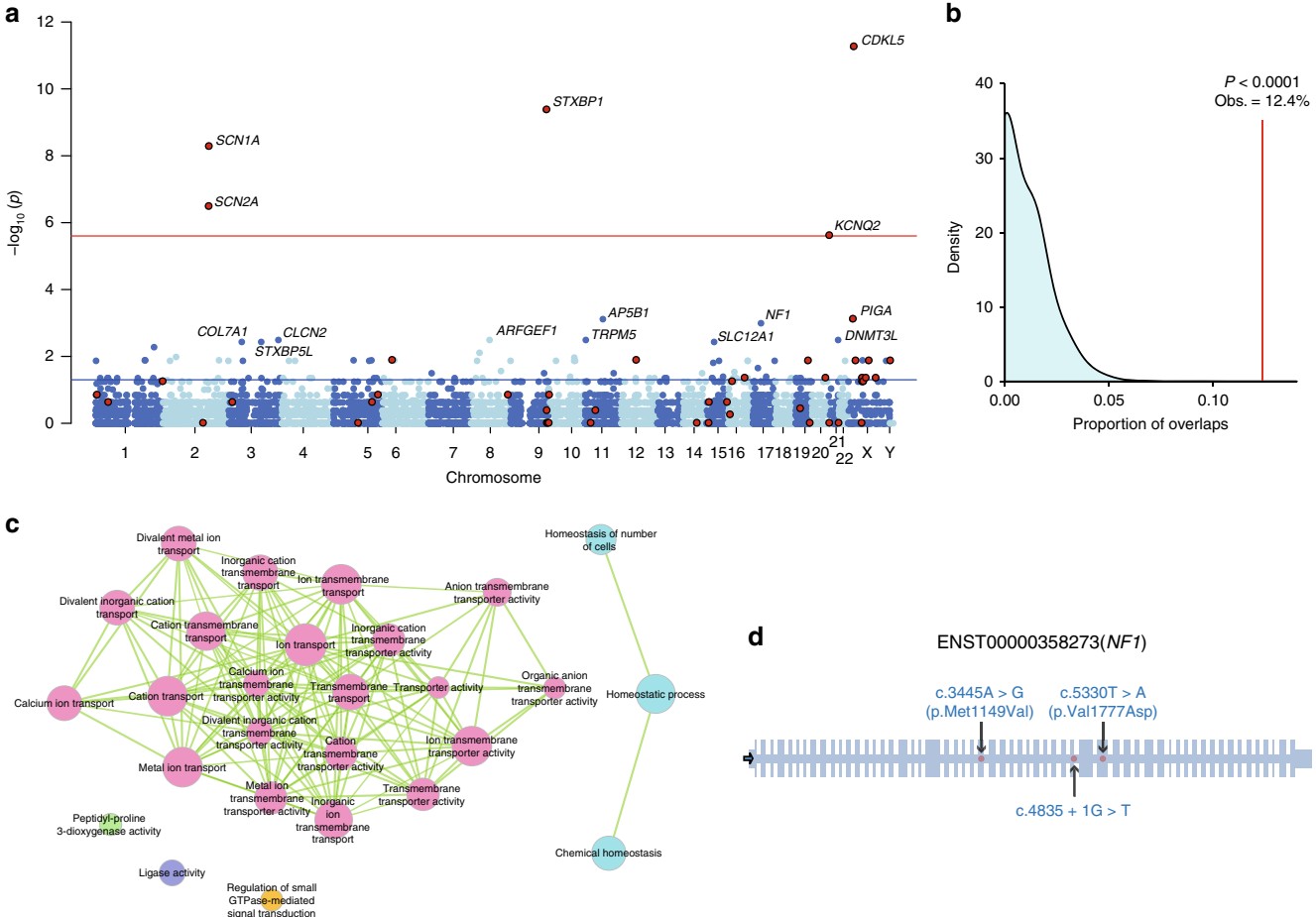

**Fig. 3** Gene-based burden test of dURVs. **a** Manhattan plot of *P* values for each gene obtained from the gene-based burden test. 58EE/DEE genes are indicated by the red-filled black circles. The red and blue horizontal lines indicate the thresholds for exome-wide significance ($P = 2.5 \times 10^{-6}$) and the nominal significance ($P = 0.05$), respectively. Genes with $P < 0.005$ are labeled with their gene symbols. **b** The observed overlap between genes with nominally significant ($2.5 \times 10^{-6} \leq P < 0.05$) dURV burden and 58EE/DEE genes (the red vertical line: 12.4% of the nominally significant genes overlapped with 58EE/DEE genes), and an expected distribution of overlaps obtained from random shuffling of the case−control labels in our cohort (cyan area). The corresponding *P* value calculated from 10,000 times of random shuffling is shown above the red vertical line. **c** Networks of gene ontology terms overrepresented among the non-58EE/DEE genes with nominally significant burden of dURVs in EE/DEE. Nodes are color-coded by clusters. Node size represents the significance of enrichment. Edge width represents the overlap coefficient. **d** A schematic representation of the exon−intron structure (adapted from ExAC Browser[10]) of *NF1* and the locations of damaging DNMs identified in our EE/DEE cohort

Table 7). Taken these together, pathogenicity of these damaging *CACNA1E* DNMs is quite convincing, and this gene should be considered as an established EE/DEE gene. Among the other tested genes, we found that three CD missense URVs in *HUWE1*, a gene primarily reported as an intellectual disability gene in literatures[30,31], are hemizygous (two maternally inherited variants and one DNM), providing additional evidence for association of this gene with EE/DEE.

**Confirmatory analyses by updating ExAC to gnomAD.** Lastly, we performed confirmatory analyses by applying a further filtering using the data of the Genome Aggregation Database (gnomAD[32], non-neuro subset of version 2.1, 104,068 exomes and 10,636 genomes). When we repeated the overall enrichment analyses of various functional types of URVs in EE/DEE (in Figs. 1 and 2, 66 tests in total), we largely replicated the findings in analyses without the gnomAD-based filtering (Supplementary Fig. 9, correlation coefficient > 0.99 for both $-\log_{10} P$ values and $\log_2$ ORs in the 66 tests in Supplementary Data 3 and 9). By repeating the gene-based burden test (in Fig. 3 and Supplementary Fig. 8), we confirmed exome-wide significant enrichment of dURVs in five known genes (*CDKL5*, *STXBP1*, *SCN1A*, *SCN2A*,

and *KCNQ2*) in EE/DEE cases as well as of damaging *NF1* DNMs in infantile spasm, while there was no gene newly reached to the significance threshold by application of the additional filtering (Supplementary Data 10). More detailed results of the analyses with an additional genomAD-based filtering are described in Supplementary Note 6.

## Discussion

In this study, we extensively analyzed WES data of EE/DEE by utilizing the case−control approach. We summarize our study by overlaying the findings onto the schema of allelic architecture of complex human diseases (Fig. 4, adapted from ref. [33]).

By comparing the profiles of URVs in cases and controls, we observed biologically interpretable patterns of URV enrichment in EE/DEE, i.e. excess of URVs predicted to be harmful to gene function such as null, CD missense and likely functional synonymous variants. This was especially true for 58EE/DEE genes, in which dURVs are more than ten times frequent in cases when compared with controls. Besides striking excess of dURVs in 58EE/DEE genes, we observed that dURVs in the other non-58EE/DEE genes are significantly enriched in EE/DEE, indicating

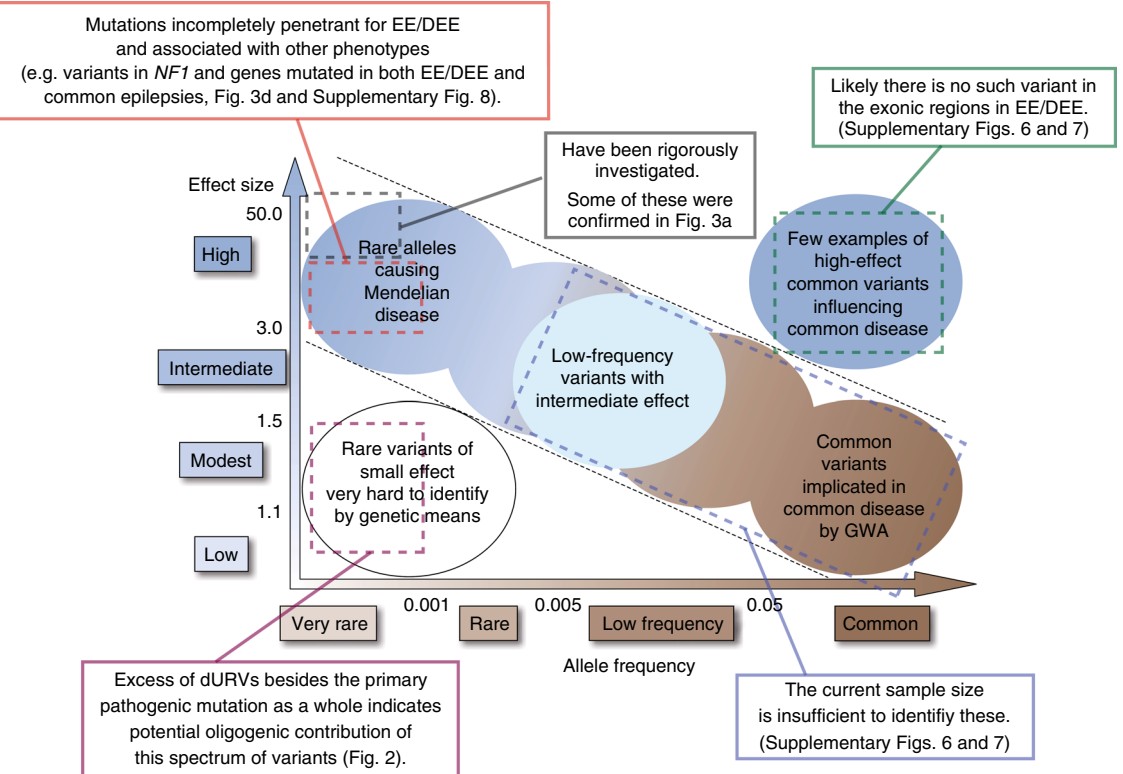

**Fig. 4** Insights into the genetic architecture of EE/DEE. Summary of the insights into the genetic architecture of EE/DEE obtained from this study. Adapted from Fig. 1 of ref. [33]. Findings in this study, corresponding spectrum of allele frequency/effect size and related figures/tables are indicated

existence of unrecognized EE/DEE genes among them. To increase the chance to discover such genes, we stratified the case cohort into individuals with and without dURVs in 58EE/DEE genes, and observed that enrichment of dURVs in non-58EE/DEE genes remains significant among the EE/DEE subset with pURVs (Fig. 2d). Though we initially considered this observation as unexpected, there is accumulating evidence pointing to the contribution of modifier/oligogenic rare variants to the etiology of neurodevelopmental disorders[34–36]. Also it was demonstrated that common variants contribute to risk of neurodevelopmental disorders even among individuals with diagnostic pathogenic variants[37,38], further suggesting a role of modulatory variants. Our data provide an additional support to this concept that is rapidly emerging but was not described specifically in EE/DEE (the magenta box in Fig. 4).

In the gene-based burden test of dURVs, we identify five genes with exome-significant dURV burden (*CDKL5*, *STXBP1*, *SCN1A*, *SCN2A* and *KCNQ2*), all of which were known EE/DEE genes. In addition, we found significant enrichment of 58EE/DEE genes as well as genes involved in known EE/DEE-related pathway (i.e. ion transportation) among the genes with nominally significant dURV burden. Therefore, these genes should be good candidates for EE/DEE genes. Among these nominally significant genes, we found exome-significant enrichment of damaging DNMs in *NF1* in infantile spasm. These three individuals with a damaging DNM in *NF1* did not harbor other variants that can explain infantile spasm. By revisiting clinical phenotypes of the three *NF1* DNM carriers (Supplementary Table 7), we found symptoms compatible with NF1 (e.g. café-au-lait spots) in two patients (carrying c.3445A > G [p.Met1149Val] or c.4835+1G > T variant; both are registered in HGMD with no clear description of epilepsy). The patient with the c.4835+1G > T variant also carries a de novo frameshift variant in *MAGEL2* [39], the gene associated with Schaaf−Yang syndrome[40], while this *MAGEL2* variant observed in the

maternal allele was unlikely to be the primary cause of the observed phenotype[39,40]. On the other hand, we did not observe features of NF1, including cutaneous and eye symptoms (e.g. Lisch nodule), in the other case (carrying c.5330T > A [p. Val1777Asp] variant; not registered in HGMD). This may perhaps suggest that some *NF1* mutations can specifically cause EE/DEE phenotypes, while there would be a possibility that symptoms suggestive of NF1 in this patient aged 7 years (in September 2018) become apparent afterward. It is known that epilepsy is relatively uncommon in patients with NF1, while the prevalence of seizures in NF1 is higher than the general population (6–10 vs. ~0.5%)[41]. The most common form of epilepsy in NF1 patients is focal seizure and EE/DEE is not frequent[42,43]. Prevalence of infantile spasm, which was observed in all *NF1* damaging DNM carriers identified in this study, in NF1 was reported as 0.54% (1/184) in ref. [44], 0.32% (2/630) in ref. [43], and 0.76% (2/260) in ref. [45]. In HGMD, there is no *NF1* variant primarily associated with an epilepsy syndrome except for a 2.8 Mb mosaic deletion encompassing *NF1* and many other genes[46]. Our result indicates that damaging *NF1* variants can be an often overlooked but relatively frequent cause of EE/DEE, especially of infantile spasm. This information would be particularly important because some of the intractable epilepsies in NF1 can be treated by surgical procedures[47]. Also, it is possible that damaging *NF1* variants are incompletely penetrant for EE/DEE. Such rare variants associated with both EE/DEE and other phenotypes with incomplete penetrance, including variants in genes commonly enriched for dURVs in EE/DEE and milder form of epilepsies (Supplementary Fig. 8), should play a certain role in the overall genetic architecture of EE/DEE (the red box in Fig. 4).

Among the EE/DEE cases with a dURV in the other genes subjected to an additional analysis (*ARFGEF1*, *STXBP5L*, *HUWE1* and *CACNA1E*), we did not observe statistically significant enrichment of a specific EE/DEE subset under the hypergeometric

distribution. However, we note that two cases with a confirmed DNM in *CACNA1E* and another case with a previously reported *CACNA1E* dURV (c.1054G > A [p.Gly352Arg]; thus, this variant fulfill the criteria for Likely Pathogenic in the ACMG guideline[48]) were all diagnosed with infantile spasm. Also, three cases with a hemizygous (two maternally inherited variants and one DNM) CD missense URV in *HUWE1* were all with infantile spasm. These observations could be at least suggestive of the phenotypic specificities in individuals carrying likely pathogenic variants in *CACNA1E* and *HUWE1*.

Considering limitations of our study, one essential thing is that the currently analyzed sample size was substantial but may not be sufficient, especially for the analysis stratifying EE/DEE cases into small subsets. Meanwhile, post hoc calculation of statistical power demonstrated that the analysis of non-58EE/DEE gene dURVs among 116 EE/DEE cases carrying pURVs in 58EE/DEE genes and 2366 controls (Fig. 2d) can achieve good statistical power when testing a limited number of hypotheses (93% for $\alpha = 0.05$, 75% for $\alpha = 0.0056$ (0.05 divided by the maximum number of tests in a panel of Figs. 1 or 2), and 52% for $\alpha = 0.00076$ (0.05 divided by the total number of tests in Figs. 1 and 2)). Therefore, enrichment of non-58EE/DEE gene dURVs among the 116 EE/DEE cases carrying pURVs, which showed statistical significance even after applying a stringent Bonferroni correction, was, to say the least, considered as a moderately reliable finding. Nevertheless, replication studies in independent cohorts are warranted to further evaluate validity of this finding, and to test if other results with nominal significance in our analyses are true or not. In addition, regarding common variants, it will require much more samples to identify common exonic SNPs with small effect sizes and low-frequency SNPs with intermediate effects (the blue box in Fig. 4), while our analysis clearly indicates that there is no common exonic SNP with a large or intermediate effect (the green box in Fig. 4). Another important point that should be considered is that specific subtypes of EE/DEE would be over- or under-represented in our EE/DEE cohort. This makes it difficult to compare the results of our and other studies. In particular, our cohort includes cases of Ohtahara or Doose syndromes, who had been recruited for the specific projects, and thereby is expected to be enriched for these subtypes. When our cohort was compared with other studies for the prevalence of EE/DEE in Japan (Okayama district)[49] or far-east Asia[50], proportions of unspecified EE/DEE (37%, 37% and 40% in our study, ref. [49] and ref. [50], respectively) and infantile spasms (36, 39 and 36%) were similar across studies. On the other hand, there were expectable over-representations of Ohtahara syndrome (12, 0 and 1%) and Doose syndrome (3, 1 and 1%) in our cohort. Early myoclonic encephalopathy (3, 1 and 0%) and migrating partial seizures of infancy (4, 0 and 0%) were also more frequent in our cases, while Lennox−Gastaut syndrome (3, 5 and 11%) was less frequent. This information should be taken into account in future meta/mega-analyses incorporating our data.

Overall, our findings highlight complexity of the genetic and phenotypic architectures of EE/DEE, which would be more complicated than what we had assumed. Specifically, unexpected enrichment of non-58EE/DEE gene dURVs in carriers of pURVs indicates roles of genetic factors other than the primary pathogenic mutations in phenotypic expressivity. These results from data-driven analyses clearly demonstrate that EE/DEE is not an aggregate of simple Mendelian disorders. Further large-scale collaborative studies will greatly aid better understanding of such complicated genetic landscape of EE/DEE.

## Methods

**Studied cohort.** We initially included 749 EE/DEE cases and 2381 controls in our study. Inclusion criteria for the case group were: (1) classified EE, such as infantile

spasm and Ohtahara syndrome (Supplementary Table 1), and (2) unspecified DEE showing the onset of epileptic seizures less than 14 years of age and developmental delay or intellectual disability. Clinical diagnoses were made by trained pediatricians/neurologists according to clinical manifestations and electroencephalogram/brain imaging findings. To describe the overall case cohort, we use the term EE/DEE. Control subjects were defined as individuals with no history of child-onset neurodevelopmental disorder. The vast majority of these individuals also have no severe non-neurological disease at the time of recruitment, while they did not receive an in-detail assessment of non-neurologic disorders (e.g. metabolic syndrome, history of cancer etc.) for this study. Written informed consent was obtained from all the participants. The study protocol was approved by the institutional review boards of Yokohama City University Faculty of Medicine.

**Exome sequencing and variant calling.** Genomic DNA was extracted from blood or saliva with standard protocols. Exome libraries were prepared by using the SureSelect Human All Exon kit (v4, 5 or 6, Agilent Technologies, Santa Clara, CA, USA). Sequencing was performed by HiSeq 2000/2500 (Illumina, San Diego, CA, USA) with 100 bp paired-end reads. Obtained reads were mapped onto human reference genome (hg19) using NovoAlign (Novocraft Technologies, v3.02), which outperforms other mapping software including BWA and BWA-MEM in mapping accuracy[51]. PCR duplicates were removed by Picard (http://picard.sourceforge.net/, v1.98). Variant calls were made by Genome Analysis Toolkit (GATK, version 3.4 and 3.7) following its best practices[52]. Briefly, de-duplicated binary alignment map (BAM) files generated by the above procedures were subjected to local realignment by RealignerTargetCreator and IndelRealigner of GATK. Quality scores were recalibrated by BaseRecalibrator and PrintReads of GATK. Genomic variant call format (gVCF) files were generated by HaplotypeCaller and then subjected to joint genotyping with GenotypeGVCFs. Obtained VCF file was subjected to variant quality score recalibration (VQSR). We included variant calls with PASS flag in the downstream analyses.

**Selection of URVs.** To extract URVs, we first excluded variants that are found in the ExAC dataset ($n = 60,706$, various ethnicities)[10], the ESP dataset ($n = 6503$, European and African Americans)[11] or the ToMMo 2K panel ($n = 2049$, Japanese)[12]. We next extracted variants only once observed in our overall cohort (749 cases and 2381 controls) using the singletons function of vcftools[53]. From the list of singleton variants, we selected (1) heterozygous calls on autosomes, (2) heterozygous calls on pseudoautosomal regions (PAR), (3) heterozygous calls on non-PAR X chromosome of female and Klinefelter individuals, (4) homozygous calls on non-PAR X chromosome of males (i.e. hemizygous variants), (5) homozygous calls on Y chromosome of male and Klinefelter individuals (i.e. hemizygous variants). We annotated these URVs using SnpEff[17] with Ensembl gene models from database GRCh37.75 activating the -canon (canonical transcripts) option. Based on the annotated information (see Supplementary Table 2), we counted the number of coding URVs in each individual. Per-individual URV counts were in the range of 23–201 (average ± SD = 54.7 ± 9.7). From the overall cohort of 3130 individuals, we excluded 21 individuals (six cases and 15 controls) carrying exceptionally small or large numbers of URVs (Smirnov−Grubbs $P < 0.001$; $n$ of URVs = 23 or 82–201) as outliers in a phenotype-blinded manner. After excluding these individuals, per-individual URV counts were approximately normally distributed (Supplementary Fig. 1, average ± SD = 54.4 ± 8.2). Among the 3109 individuals (743 cases and 2366 controls) included in the downstream analyses, there were 169,014 and 42,974 URVs in coding and noncoding regions, respectively.

**Principal component analysis.** Principal component analysis was performed by using information of common exonic SNPs. For this analysis, we extracted variants that are: (1) on the InfiniumExome-24v1-0_A1 genotyping array, (2) with MAF > 5% in East Asian (EAS) population of ExAC[10] and (3) biallelic in EAS ($n$ of SNPs = 12,173). We then combined information of these SNPs in our cohort and the data of the same SNPs in European American (CEU: Utah residents with Northern and Western European ancestry from the CEPH collection), Japanese (JPT: Japanese in Tokyo, Japan), Han-Chinese (CHB: Han Chinese in Beijing) and African (YRI: Yoruba in Ibadan, Nigeria) individuals of the 1000 Genomes Project[54]. The combined dataset of common SNPs was subjected to further filtering and linkage disequilibrium (LD)-based pruning using PLINK[55,56], with the following options and parameters: –maf (minimum minor allele frequency) 0.1, –mind (maximum per-person missing rate) 0.1, –geno (maximum per-SNP missing rate) 0.1, –hwe (Hardy−Weinberg disequilibrium $P$-value) 0.001 and –indep (SNP window size, number of SNPs to shift and variance inflation factor threshold) 50 5 2. By using information of the SNPs that passed these filters, we performed a principal component analysis with PLINK[55,56]. According to the results of these analyses, we confirmed that the individuals included in this study form a single cluster (Supplementary Fig. 2). We incorporated the first ten principal components into our regression analysis as described below.

**Additional quality controls (QCs).** Sex of the individuals was confirmed by –check-sex ycount function of PLINK[55,56] (Supplementary Fig. 3). For this analysis, we first extracted variant calls within the coding regions (defined by RefSeq) of non-PAR X and Y chromosomes. Then individual-level genotype calls with read

depth <10 were masked by the VariantFiltration function of GATK enabling the −setFilteredGtToNocall option. From this list of variant calls, we extracted 1121 SNPs with MAF > 1% and missing genotype rate <10% and subjected them to the −check-sex ycount analysis. Through this analysis, we identified two EE/DEE cases with 47,XXY karyotype (Klinefelter syndrome), in accordance with the clinical information. Regarding relationship among the studied 3109 subjects, no pair was with PI_HAT (proportion of identity by descent, calculated by PLINK) >0.2, and 0.0001% (5/4,831,386) and 0.058% (2,812/4,831,386) of the pairs were with PI_HAT >0.15 and >0.1, respectively. Exome coverage in each individual was estimated by the CallableLoci function of GATK with default parameters. Average (±SD) of the proportions of callable loci (regions marked with CALLABLE flag) among the RefSeq-defined coding regions were 96.77 (±0.74)% in 743 cases and 96.64 (±0.63)% in 2366 controls. We incorporated these proportions of callable loci into our regression analysis as described below.

**Logistic regression analysis of URVs.** We tested for association between per-individual counts of each functional type of URVs and the case−control status by logistic regression analysis. In this analysis, we incorporated sex, proportion of callable loci and the first ten principal components calculated above as covariates. For functional classification of URVs, we initially categorized them into the following four types based on SnpEff annotation[17] (see Supplementary Table 2): null, Moderate (mainly missense and inframe indel), synonymous, and noncoding URVs. Note that we reclassified protein−protein contact and start-lost variants, which were originally included in the High impact group according to the SnpEff annotation, into the Moderate group. We then extracted subsets of Moderate and synonymous URVs more likely to be functional (i.e. consensus-damaging missense URVs (CD missense URVs), synonymous URVs within DNase I hypersensitive sites in frontal cortex (FCDHS synonymous URVs), and synonymous URVs in splice regions (SR synonymous URVs)). By following the method described in ref. [14], we defined CD missense URVs as those predicted to be damaging by all of the following seven algorithms: SIFT[57], PolyPhen-2[58] (HumVar and HumDiv models), LRT[59], MutationTaster[60], Mutation Assessor[61] and PROVEAN[62] (these annotations were added by dbNSFP3.0a[63]). FCDHS synonymous URVs were defined as those within the DNase I hypersensitive sites in Cerbrm_frnt_Pk dataset downloaded from the UCSC Table Browser, by following the method described in ref. [19]. SR synonymous URVs were defined as those at the last and the first 3 bp of an exon adjacent to an intron, which were extracted by using the splice region annotation by SnpEff[17]. Based on these classifications, we analyzed the following nine variant categories: null, Moderate, synonymous, noncoding, CD missense, other Moderate, FCDHS synonymous, SR synonymous, and other synonymous.

**Gene sets (high pLI genes and 58EE/DEE genes).** Genes depleted for null variants in the general population, defined as having probability-of-being-LOF-intolerant (pLI) > 0.9 (n of genes = 3230), were extracted by using the information of pLI in ExAC[10]. The set of established autosomal dominant or X-linked EE/DEE genes (n of genes = 58, referred to as 58EE/DEE genes) was compiled by combining the following gene lists: (1) established epilepsy genes described in Table S5 of ref. [20] (genes without sufficient evidence for association with EE were not included), (2) established EE genes described in Supplementary Table 3 of ref. [9], and (3) three additional genes (ATP1A3, KIAA2022 (NEXMIF) and SMC1A) selected by our manual literature/database search.

**Extraction of pURVs in 58EE/DEE genes.** In total, there were 138 EE/DEE individuals carrying null or CD missense URVs (damaging URVs (dURVs) in the main text) in 58EE/DEE genes. We tested inheritance patterns of these dURVs by Sanger sequencing whenever the DNA samples of their parents were available. We considered the following dURVs as convincingly pathogenic URVs (pURVs in the main text): (1) dURVs in autosomal 58EE/DEE genes confirmed to be de novo (including dURVs found in a mosaic status in one of the parents) (n = 77), (2) dURVs in PIGA, ARX, IQSEC2 or ARHGEF9 (X-linked recessive genes or genes whose heterozygous mutations cause a mild phenotype) in a hemizygous status (either transmitted from the mother or generated as de novo) (n = 8), (3) dURVs in PCDH19 (X-linked gene whose heterozygous mutations are specifically pathogenic) in a heterozygous status (either transmitted from the father or generated as de novo) (n = 3), and dURVs in the other X chromosome 58EE/DEE genes confirmed to be de novo (n = 28) (Supplementary Data 2). Note that dURVs in non-58EE/DEE genes that were not considered as pURV (138−116 = 22 variants) were not excluded due to a failure in Sanger validation, but because of their inheritance from one of the unaffected parents (in the cases of autosomal URVs), lack of DNA samples of the parent, or mosaicism. Reported phenotypes in the Human Gene Mutation Database (HGMD, professional version 2017.3) were annotated by requiring complete match of the reference and variant alleles. Classification of pURVs according to the reported phenotypes in HGMD was performed as shown in Supplementary Table 4 in a manner blinded to information of the other URVs and the clinical phenotypes of pURV carriers. We included specific clinically defined neurodevelopmental disorders with a strong genetic component (e.g. Rett syndrome) in the EE/DEE group, and gene-defined syndromes with no detailed information of clinical manifestation (e.g. STXBP1 encephalopathy) in the

unknown/uncertain group. We scrutinized the original publications reporting the mutation when needed.

**Multiple testing correction.** In the analyses related to Figs. 1 and 2, we tested a total of 66 hypotheses. According to this number, we performed multiple testing correction with Bonferroni and Benjamini−Hochberg procedures using p.adjust function of R.

**Gene expression properties of potential modifier dURVs.** To explore properties of non-58EE/DEE gene dURVs among the 116 pURV carriers, we first constructed[18] lists of genes specifically expressed in each of the 53 tissue/cell types in the Genotype-Tissue Expression (GTEx) project[21], using the pSI (specificity index probability) package[64] of R. We then analyzed if the case−control status (116 EE/DEE cases with pURV in 58EE/DEE genes or 2366 controls) was associated with the number of non-58EE/DEE gene dURVs in genes specific to each tissue/cell type, using logistic regression analysis incorporating sex, proportion of callable loci and the first ten principal components as covariates. In other words, we performed analyses treating genes specific to each tissue as a gene set, like the sets of intolerant genes and 58EE/DEE genes analyzed in Fig. 1b, c. For an analysis contrasting 13 brain regions with the other 40 tissues, we first sorted the tissues according to their P values obtained from logistic regression in the following order: (1) tissues with the smallest P values for dURV enrichment in EE/DEE (i.e. OR > 1), (2) tissues with the largest P values for dURV enrichment in EE/DEE, (3) tissues with the largest P values for dURV depletion in EE/DEE (i.e. OR < 1), and (4) tissues with the smallest P values for dURV depletion in EE/DEE. We then analyzed the sorted ranks for the group of 13 brain regions and the group of the other 40 tissues using two-tailed Wilcoxon rank sum test. We repeated the above analysis using lists of genes with moderate to high expression in each tissue (transcripts per million reads (TPM) > 10 in the GTEx Analysis V7 dataset), instead of the lists of specifically expressed genes, for confirmation.

**Analysis of doubleton and tripleton rare variants.** To extract doubleton and tripleton rare variants, we first excluded all variants that are found in any of ExAC[10], ESP[11] or ToMMo[12] databases. We next selected variants that were observed twice and three times in our overall cohort (749 cases and 2381 controls) as doubleton and tripleton rare variants, respectively. By following the methods used for singleton URVs, we further selected (1) heterozygous calls on autosomes, (2) heterozygous calls on PAR, (3) heterozygous calls on non-PAR X chromosome of female and Klinefelter individuals, (4) homozygous calls on non-PAR X chromosome of males (i.e. hemizygous variants), (5) homozygous calls on Y chromosome of male and Klinefelter individuals (i.e. hemizygous variants), and removed 21 outlier individuals regarding the numbers of singleton URVs. After that, there were 45,832 doubleton and 20,786 tripleton rare variants. Association of each functional type of doubleton/tripleton rare variants in a geneset and the case−control status was analyzed by the method used for singleton URVs. For the analysis of tripleton CD missense variants in 58EE/DEE genes, which were exclusively found in EE/DEE cases (i.e. in a condition of complete separation), we artificially added one variant to the control group and calculated the P value as 0.00102, and then described the P value for this analysis as P < 0.00102.

**Exome-wide association study of SNPs.** To ensure quality of the single nucleotide polymorphism (SNP: defined as variants with MAF > 1%, including small indels) calls, we first masked genotype calls with read depth < 10 using the VariantFiltration function of GATK enabling the −setFilteredGtToNocall option. We then extract SNP calls with PASS flag and subjected them to filtering by PLINK[55,56] with the following options and parameters: −maf 0.01, −geno 0.1, −hwe 0.00001 −biallelic-only and −indep 50 5 2. After filtering, there was a total of 35,375 SNPs. Association of individual SNPs with EE/DEE was tested by a logistic regression analysis incorporating sex, proportion of callable loci and the first ten principal components as covariates. Quantile−quantile plot was generated by qqman package[65] of R. Statistical power obtained from our cohort was calculated by pwr.2p2n.test function in pwr package of R. Genomic inflation factor (lambda) was calculated by GenABEL package[66] of R with the default regression method. Proportion of the phenotypic variance explained by these exonic SNPs was estimated by the Genomic Restricted Maximum Likelihood method in the Genome-wide Complex Trait Analysis toolkit (GCTA-GREML)[67].

**Gene-based burden test of dURVs.** According to the results of our logistic regression analysis testing relationship between URVs and EE/DEE, we included dURVs (null and CD missense) in the gene-based burden test. Analysis was performed by using collapsing method[22] followed by two-tailed Fisher's exact test comparing the proportions of cases and controls carrying one or more dURV in a gene. After performing an analysis including all genes and individuals, we repeated the analysis by excluding 116 cases carrying pURVs from the sample list and 58EE/DEE genes from the gene list. Manhattan plot of the obtained result was generated by using qqman package[65] of R.

**Overlap of nominally significant genes with 58EE/DEE genes**. To evaluate probability for our observation of overlap between genes with nominally significant ($2.5 \times 10^{-6} \leq P < 0.05$) dURV burden and 58EE/DEE genes, we performed an analysis based on random shuffling of the case−controls status. In practice, we first repeatedly shuffled case−control labels of our cohort, performed burden tests with the shuffled labels, and generated 10,000 sets of nominally significant genes. The numbers of nominally significant genes in these 10,000 sets were in the range 41–108. For each set of nominally significant genes generated by random shuffling, we calculated the proportion of genes overlapped with the 58EE/DEE genes over the total gene count in a set of nominally significant genes. We then estimated probability for our observation (12 of 97 nominally significant genes were in the list of 58EE/DEE genes in the real dataset) as the fraction of iterations where the proportion of overlapping genes was equal to or greater than what we have observed in the real dataset.

**GO enrichment analysis**. Gene ontology (GO) enrichment analysis of non-58EE/DEE genes with nominally significant dURVs burden in EE/DEE was performed by using ToppGene[23], with the default parameters. Visualization of the clusters of significantly enriched GO terms was performed by connecting pairs of nodes sharing the containing genes with an overlap coefficient >0.8 using Enrichment-Map plugin of CytoScape (http://apps.cytoscape.org/apps/enrichmentmap).

**Statistical assessment of DNMs in selected genes**. Based on nominally significant excess of dURVs in our gene-based burden test excluding pURV carriers ($n$ of individuals = 627), and extremely high intolerance against null variants in the general population (pLI score > 0.99), we selected five candidate genes (*NF1*, *ARFGEF1*, *STXBP5L*, *HUWE1*, and *CACNA1E*). We analyzed inheritance patterns of dURVs in these genes by Sanger sequencing whenever the DNA samples of the parents were available. Probability of identifying the observed number of, or more damaging DNMs in a selected gene was calculated on an established model of per-gene mutation rate[28]. Based on this model, we calculated per-haploid null DNM rate ($\mu_{null}$) as the sum of the rates for nonsense, frameshift and splice site DNMs, and per-haploid CD missense DNM rate ($\mu_{CDmissense}$) as the rate for all missense DNMs multiplied with 203/1252, that is, the proportion of CD missense DNMs over the all missense DNMs observed in the control cohorts (unaffected siblings) in studies of autism spectrum disorders analyzing a total of ~2000 control trios[16,68,69]. Based on these DNM rates, we estimated the expected number of null/CD missense DNMs in a gene as ($\mu_{null} + \mu_{CDmissense}$) × 2 × 627, and then calculated the probability of identifying the observed number of, or more damaging DNMs in our cohort under the Poisson distribution.

**Comparison with common epilepsies**. The results of gene-based burden tests for common epilepsies (GGE and NAFE) were obtained from Tables S10−12 of ref. [20]. For NAFE, we combined the data of familial and sporadic NAFE in Tables S11−12 of ref. [20] (in each table familial or sporadic NAFE were compared with the shared controls) and obtained a single $P$ value for each gene by two-tailed Fisher's exact test, with the following 2 × 2 table: rows, cases (familial and sporadic NAFE) and controls; columns, the numbers of individuals with and without qualifying variants described in ref. [20]. 3D-plotting figure was generated by using plot3D package of R. Note that selection criteria for rare variants in our study (URVs) and the study for common epilepsy (qualifying variants in the corresponding publication) were not identical. Analysis of overlaps of nominally significant genes ($P < 0.05$) between EE/DEE and GGE or NAFE was performed by randomly shuffling the case−control status of our cohort as described in the "Overlap of nominally significant genes with 58EE/DEE genes" section above.

**Post hoc statistical power calculation**. We performed post hoc calculation of statistical power for detection of non-58EE/DEE gene dURV enrichment in 116 EE/DEE cases carrying pURVs in 58EE/DEE genes using G*Power[70]. The parameters used were: test type = logistic regression; tail(s) = two-tails, OR at one standard deviation above the mean = 1.380, standard deviation of per-individual non-58EE/DEE gene dURV count = 2.840, baseline probability $\Pr(Y = 1|X = 1)$ H0 = 0.0467 (116/2,482), total sample size = 2482, $R^2$ of the other covariates = 0.00283, $X$ distribution = normal. Standard deviation of per-individual non-58EE/DEE gene dURV count was calculated as:

$$\sqrt{\frac{(116 - 1)\mathrm{SD}^2_{case} + (2366 - 1)\mathrm{SD}^2_{control}}{116 + 2366 - 2}}.$$

$R^2$ of the other covariates (sex, proportion of callable loci and the first ten principal components) was calculated by lm function of R.

**Confirmatory analyses using the gnomAD data**. We repeated the enrichment analyses of various functional types of URVs in EE/DEE (in Figs. 1 and 2) and the gene-based burden test (in Fig. 3 and Supplementary Fig. 8) using the data of URVs to which a further filtering based on the gnomAD dataset (version 2.1)[32] was applied. In this analysis, we used the non-neuro dataset of gnomAD, which include only samples from individuals who were not ascertained for having a neurological condition in a neurological case−control study (104,068 exomes and 10,636

genomes in total). By applying this additional filtering, 24,246/169,014 (14.3%) coding and 5595/42,974 (13.0%) noncoding URVs were removed.

**Reporting summary**. Further information on research design is available in the Nature Research Reporting Summary linked to this article.

## Data availability

Exome data will be available in the Human Genetic Variation Database (http://www.hgvd.genome.med.kyoto-u.ac.jp/repository/HGV0000009.html). Note that raw sequence and individual-level genotype data can be provided via formal collaboration due to the contents of the obtained informed consent. All other data are contained within the article and its supplementary information or upon reasonable request from the corresponding authors.

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

## Acknowledgements

We thank all the study participants and members of the DEEPEN (DEvelopmental and EPileptic ENcephalopathy) Consortium (the full list of consortium members is provided in Supplementary Note 7). This work was supported in part by a grant for Research on Measures for Intractable Diseases, a grant for Comprehensive Research on Disability, Health and Welfare, the Strategic Research Program for Brain Science (SRPBS) from the Japan Agency for Medical Research and Development (AMED) (grant numbers: JP18ek0109280, JP18dm0107090, JP18ek0109301, JP18ek0109348, JP18dm0107133, JP18ek0109381 and JP18kk020500); a Grant-in-Aid for Scientific Research on Innovative Areas (Transcription Cycle) from the Ministry of Education, Culture, Sports, Science and Technology of Japan (MEXT); Grants-in-Aid for Scientific Research (A, B, and C), for Young Scientists (A) and for Challenging Exploratory Research from the Japan Society for the Promotion of Science (JSPS) (grant numbers: JP17H01539, JP16H05160, JP16H05357, JP16H06254, JP15K10367, JP17K10080, JP 17K15630 and JP17H06994); the fund for Creation of Innovation Centers for Advanced Interdisciplinary Research Areas Program in the Project for Developing Innovation Systems from the Japan Science and Technology Agency (JST); the Takeda Science Foundation; grants from the Ministry of Health, Labour and Welfare of Japan; the Yokohama Foundation for Advancement of Medical Science; the Hayashi Memorial Foundation for Female Natural Scientists; The Ichiro Kanehara Foundation for the Promotion of Medical Science & Medical Care; and The Japan Epilepsy Research Foundation.

## Author contributions

Conceptualization, A.T. and N.Matsumoto; Methodology, A.T.; Software, A.T.; Validation, A.T., M.N., H.S., T.M., S.Mitsuhashi, N.Miyake, S.Miyatake, N.T., K.Iwama, G.M., F.S., A.F., E.I., E.K., Y.U., K.Hamanaka, C.O., T.I., H.A., K.S., T.Sakaguchi and K.D.; Formal analysis, A.T., M.N., H.S., T.M., S.Mitsuhashi, N.Miyake, S.Miyatake, N.T., K. Iwama, G.M., F.S., A.F., E.I., E.K., Y.U., K.Hamanaka, C.O., T.I., H.A., K.S., T. Sakaguchi

and K.D.; Investigation, A.T., M.N., H.S., T.M., S.Mitsuhashi, N.Miyake, S.Miyatake, N.T., K.Iwama, G.M., F.S., A.F., E.I., E.K., Y.U., K.Hamanaka, C.O., T.I., H.A., K.S., T.Sakaguchi, K.D., R.Takahashi and M.Kato; Resources, Y.T., N.O., H.O., K.N., J.Tohyama, K.Haginoya, S.T., I.K., T.Okanishi, T.G., M.Sasaki, Y.S., H.I., T.Y., K.T., S.Y., T.Oboshi, K.Imai, T.K., Y.K., M.Kubota, H.K., S.B., M.I., R.K., M.H., M.O., Y.M., R.M., J.Takanashi, J.M., K.Y., M.Shimono, M.A., R.Takayama, S.H., K.A., H.M., S.N., T.Shii-hara, M.Kato; Data curation, A.T., M.N., H.S., T.M., S.Mitsuhashi, N.Miyake, S.Miyatake, N.T., K.Iwama, G.M., F.S., A.F., E.I., E.K., Y.U., K.Hamanaka, C.O., T.I., H.A., K.S., T.Sakaguchi, K.D., R.Takahashi and M.Kato; Writing—original draft, A.T.; Writing—review and editing, A.T., M.N., H.S., T.M., S.Mitsuhashi, Y.T., N.O., H.O., K.N., J.Tohyama, K.Haginoya, S.T., I.K., T.Okanishi, T.G., M.Sasaki, Y.S., N.Miyake, S.Miya-take, N.T., K. Iwama, G.M., F.S., A.F., E.I., E.K., Y.U., K.Hamanaka, C.O., T.I., H.A., K.S., T.Sakaguchi, KD, R.Takahashi, H.I., T.Y., K.T., S.Y., T.Oboshi, K.Imai, T.K., Y.K., M.Kubota, H.K., S.B., M.I., R.K., M.H., M.O., Y.M., R.M., J.Takanashi, J.M., K.Y., M.Shimono, M.A., R.Takayama, S.H., K.A., H.M., S.N., T.Shiihara, M.Kato and N.Matsumoto; Visualization, A.T.; Supervision, N.Matsumoto; Project administration, M.Kato and N.Matsumoto; Funding acquisition, A.T., M.N., H.S., T.M., S.Mitsuhashi, N.Miyake, S.Miyatake, M.Kato and N.Matsumoto. All authors read and approved the final manuscript before submission.

## Additional information

**Competing interests:** The authors declare no competing interests.

Atsushi Takata[1,34], Mitsuko Nakashima[1,2,34], Hirotomo Saitsu[1,2,34], Takeshi Mizuguchi[1], Satomi Mitsuhashi[1], Yukitoshi Takahashi[3], Nobuhiko Okamoto[4], Hitoshi Osaka[5], Kazuyuki Nakamura[6], Jun Tohyama[7], Kazuhiro Haginoya[8], Saoko Takeshita[9], Ichiro Kuki[10], Tohru Okanishi[11], Tomohide Goto[12], Masayuki Sasaki[13], Yasunari Sakai[14], Noriko Miyake[1], Satoko Miyatake[1], Naomi Tsuchida[1], Kazuhiro Iwama[1], Gaku Minase[1], Futoshi Sekiguchi[1], Atsushi Fujita[1], Eri Imagawa[1], Eriko Koshimizu[1], Yuri Uchiyama[1], Kohei Hamanaka[1], Chihiro Ohba[1], Toshiyuki Itai[1], Hiromi Aoi[1], Ken Saida[1], Tomohiro Sakaguchi[1], Kouhei Den[1], Rina Takahashi[1], Hiroko Ikeda[3], Tokito Yamaguchi[3], Kazuki Tsukamoto[3], Shinsaku Yoshitomi[3], Taikan Oboshi[3], Katsumi Imai[3], Tomokazu Kimizu[15], Yu Kobayashi[7], Masaya Kubota[16], Hirofumi Kashii[16], Shimpei Baba[11], Mizue Iai[12], Ryutaro Kira[17], Munetsugu Hara[18], Masayasu Ohta[19], Yohane Miyata[20], Rie Miyata[21], Jun-ichi Takanashi[22], Jun Matsui[23], Kenji Yokochi[24], Masayuki Shimono[25], Masano Amamoto[26], Rumiko Takayama[27], Shinichi Hirabayashi[28], Kaori Aiba[29], Hiroshi Matsumoto[30], Shin Nabatame[31], Takashi Shiihara[32], Mitsuhiro Kato[6,33] & Naomichi Matsumoto[1]

[1]Department of Human Genetics, Yokohama City University Graduate School of Medicine, 3-9 Fukuura, Kanazawa-ku, Yokohama 236-0004, Japan. [2]Department of Biochemistry, Hamamatsu University School of Medicine, 1-20-1 Handayama, Higashi-ku, Hamamatsu 431-3192, Japan. [3]National Epilepsy Center, NHO Shizuoka Institute of Epilepsy and Neurological Disorders, 886 Urushiyama, Aoi-ku, Shizuoka 420-8688, Japan. [4]Department of Medical Genetics, Osaka Women's and Children's Hospital, 840 Murodo-cho, Izumi, Osaka 594-1101, Japan. [5]Department of Pediatrics, Jichi Medical University, 3311-1 Yakushiji, Shimotsuke 329-0498, Japan. [6]Department of Pediatrics, Yamagata University Faculty of Medicine, 2-2-2 Iida-nishi, Yamagata 990-9585, Japan. [7]Department of Child Neurology, NHO Nishiniigata Chuo Hospital, 1-14-1 Masago, Nishi-ku, Niigata 950-2085, Japan. [8]Department of Pediatric Neurology, Miyagi Children's Hospital, 4-3-17 Ochiai, Aoba-ku, Sendai 989-3126, Japan. [9]Department of Pediatrics, Yokohama City University Medical Center, 4-57 Urafunecho, Minami-ku, Yokohama 232-0024, Japan. [10]Department of Pediatric Neurology, Osaka City General Hospital, 2-13-22 Miyakojimahondori, Miyakojima-ku, Osaka 534-0021, Japan. [11]Department of Child Neurology, Comprehensive Epilepsy Center, Seirei Hamamatsu General Hospital, 2-12-12 Sumiyoshi, Naka-ku, Hamamatsu 430-8558, Japan. [12]Division of Neurology, Kanagawa Children's Medical Center, 2-138-4 Mutsukawa, Minami-ku, Yokohama 232-8555, Japan. [13]Department of Child Neurology, National Center of Neurology and Psychiatry, 4-1-1 Ogawahigashi, Kodaira 187-8551, Japan. [14]Department of Pediatrics, Graduate School of Medical Sciences, Kyushu University, 3-1-1 Maidashi, Higashi-ku, Fukuoka 812-8582, Japan. [15]Department of Pediatric Neurology, Osaka Women's and Children's Hospital, 840 Murodo-cho, Izumi, Osaka 594-1101, Japan. [16]Division of Neurology, National Center for Child Health and Development, 2-10-1 Okura, Setagaya-ku, Tokyo 157-8535, Japan. [17]Department of Pediatric Neurology, Fukuoka Children's Hospital, 5-1-1 Kashiiteriha, Higashi-ku, Fukuoka 813-0017, Japan. [18]Department of Pediatrics and Child Health, Kurume University School of Medicine, 67 Asahi-machi, Kurume, Fukuoka 830-0011, Japan. [19]Department of Neuropediatrics, Aiseikai Memorial Ibaraki Welfare Medical Center, 1872-1 Motoyoshida-cho, Mito 310-0836, Japan. [20]Department of Neuropediatrics, Tokyo Metropolitan Neurological Hospital, 2-6-1 Musashidai, Fuchu 183-0042, Japan. [21]Department of Pediatrics, Tokyo-kita Medical Center, 4-17-56 Akabanedai, Kita-ku, Tokyo 115-0053, Japan. [22]Department of Pediatrics, Tokyo Women's Medical University Yachiyo Medical Center, 477-96 Owadashinden, Yachiyo 276-8524, Japan. [23]Department of Pediatrics, Shiga University of Medical Science, Setatsukinowacho, Otsu 520-2192, Japan. [24]Department of Pediatric Neurology, Seirei-Mikatahara

General Hospital, 3453 Mikatahara-cho, Kita-ku, Hamamatsu 431-1304, Japan. [25]Department of Pediatrics, School of Medicine, University of Occupational and Environmental Health, 1-1 Iseigaoka, Yahatanishi-ku, Kitakyushu 807-8555, Japan. [26]Kutakyushu Municipal Yahata Hospital Pediatric Emergency Center, 4-18-1 Nishihonmachi, Yahatahigashi-ku, Kutakyushu 805-8534, Japan. [27]Hokkaido Medical Center for Child Health and Rehabilitation, 1-240-6 Kanayama 1-jo, Teine-ku, Sapporo 006-0041, Japan. [28]Division of Neurology, Nagano Children's Hospital, 3100 Toyoshina, Azumino 399-8288, Japan. [29]Department of Pediatrics, Toyohashi Municipal Hospital, 50 Aza Hachiken Nishi, Aotake-Cho, Toyohashi 441-8570, Japan. [30]Department of Pediatrics, National Defense Medical College, 3-2 Namiki, Tokorozawa 359-8513, Japan. [31]Department of Pediatrics, Graduate School of Medicine, Osaka University, 2-2 Yamadaoka, Suita 565-0871, Japan. [32]Department of Neurology, Gunma Children's Medical Center, 779 Shimohakoda, Hokkitsu-machi, Shibukawa 377-8577, Japan. [33]Department of Pediatrics, Showa University School of Medicine, 1-5-8 Hatanodai, Shinagawa-ku, Tokyo 142-8666, Japan. [34]These authors contributed equally: Atsushi Takata, Mitsuko Nakashima, Hirotomo Saitsu.

