## [Peer Review File · Nature Communications]

Reviewers' comments:

Reviewer #1 (Remarks to the Author):

Takata and colleagues describe comprehensive exome-wide statistical investigations on 743 DEE cases compared to 2366 controls as well as to public databases. The approach and the calculations appear to be sound and extensive. However, I see some issues in the study design that need clarification or correction.

1) The authors describe 743 cases with DEE. In the supplement, these cases are sub-divided into Infantile spasm, Ohtahara, Lennox Gastaut syndrome etc. of each a few hundred or dozens of individuals. Almost all these disease entities exhibit classic Epileptic Encephalopathies (EE). According to the most recent nomenclature of the ILAE, the term "Developmental and Epileptic Encephalopathies" (DEE) is used for disorders where developmental aspects occur prior to seizure onset. DEE is not meant to describe the whole group of EE and developmental disorders with epilepsy. Thus, "DEE" is a term on the same level as "EE" and "DEE" does not include "EE". I personally find this terminology unfortunate; however, the terms EE and DEE should be used accordingly throughout the manuscript (or be exchanged by another term).

All 743 cases are of Japanese origin and are compared to 2,366 controls of Japanese origin as well as to ExAC (60,706 individuals), ESP (6,500 individuals) and ToMMo (1,070 individuals).

2) Please provide more information on the 2,366 Japanese controls. Are they healthy individuals? Any other non-neurologic disorders? Relations?

3) ExAC has been superseded by gnomAD (141,456 individuals, including all ExAC individuals with high sequencing quality). Thus, cases should be compared to gnomAD (with more individuals enabling better statistical power) and not ExAC.

4) ESP is a rather old database of minor quality compared to gnomAD. ESP could thus be omitted from the study.

5) The most important control cohort appears to be the (not further described) series of 2,366 as well as ToMMo as both comprise exclusively Japanese controls. Comparison of 743 cases versus 2,366 + 1,070 Japanese controls does not allow for robust statistical calculations. Therefore, despite ExAC or gnomAD, all ultra-rare variants found in 743 cases can very well be simply rare Japanese variants and not be disease-associated at all. Even including gnomAD would only marginally solve this problem as it includes only 152 Japanese alleles. How do the authors make sure that rare variants are not simply population-specific?

6) To evaluate the previous question – how many ultra-rare variants are found in the Japanese controls that are not reported in public databases?

- 7) How significant would be the opposite calculation – comparing ultra-rare variants in databases versus Japanese cases?
- 8) Multiple testing for 66 hypotheses appears to be low and should be considerably increased.
- 9) According to ACMG, “null variants” should be preferred over “loss-of-function variants”.
- 10) “Second-hit” effects is possibly not, what the authors intend to say. I believe “oligogenic” effects would be more suitable, is that right?

Reviewer #2 (Remarks to the Author):

In the study by Takata et. al. they perform Ultra Rare Variant (URV) case control analysis in a cohort of 743 Developmental and epileptic encephalopathy (DEE) cases. They found a significant excess of damaging URV within known DEE genes, which is not surprising. They also observed an excess of damaging URV in non-DEE genes. Gene-based burden analysis was performed with five known DEE genes being exome-wide significant. The NF1 gene which was nominally significant was a novel finding and 3 variants were validated as de-novo and upon re-analysis was found to be significant.

General comment

This manuscript was very difficult to read as it was verbose and was mixed with expected results and potentially interesting results. I found that the expected and therefore uninteresting results took a disproportionate amount of space, leaving the reader frustrated that they have to search very hard for the interesting results. For example, the discovery of excess of damaging variants in known DEE genes is expected. The gene-based burden test showing five known DEE genes is an expected result.

Comments

1. The excess of damaging URV in non-DEE genes is an interesting and novel result. However, there is little explanation in the results and discussion on what genes make up this gene list. It is buried in the methods and currently is difficult to interpret how it was generated. Also, how were the significance thresholds determined for tests involving non-DEE genes?
2. If there is nothing interesting, the analysis of doubletons and triplettons should be moved to the supplementary and condensed to a 1-2 lines in the main text instead of being 1.5 pages.

3. Similarly, exome-wide association of SNPs should be moved to the supplementary as there are no interesting results and shouldn't consume 0.5 pages.

4. Lastly, the same applies for "Comparison between DEE and common epilepsies". The results of the analysis is nominally significant and the novel result of de novo variants in NF1 would be more interesting then ending with another uninteresting result.

5. The above should reduce the large size of the abstract that's littered with uninteresting results and unimportant conclusions.

5. In the first paragraph of the discussion, the comments of doing a separate analysis instead of contributing to a better powered consortium analysis I don't agree with entirely. However, I would like the authors to expand on the last comment, "we conducted more various statistical analyses to better understand the genetic architecture of DEE" and be explicit on the exact analysis and why this would not have been possible in the Zhu et al. study?

6. Apart from exome-wide significance, throughout the manuscript it is difficult to know what the threshold for significance after adjusting for multiple test correction. It is not explicitly stated in the results text or indicated on the figures. Futhermore, there should be an indication of significance on the figures.

7. A comment or discussion should be made on the odds ratio and how this compares with previous results. This makes up the bulk of the other dimension (i.e. x-axis) of the data presented in Figures 1 and 2.

Reviewer #3 (Remarks to the Author):

Takata et al report a comprehensive genetic analysis study designed to understand the genetic architecture of developmental and epileptic encephalopathy (DEE). This is a very large cohort study of a severe disabling disorder from a single population. An impressive cohort of 743 DEE cases and 2,366 healthy individuals from Japan received exome analysis. The analysis strategy focused on ultra-rare variants (URVs) observed only once in this cohort and not seen in public databases The key

findings were: (i) damaging (LOF or deleterious missense) URVs were significantly enriched in DEE cases, in both known DEE and non-DEE genes, (ii) enrichment in non-DEE genes remained even after removal of a subset of cases with convincingly pathogenic variants suggesting “second hit” effects in non-DEE genes, (iii) significant burden of damaging URVs in five known DEE genes (CDKL5, STXBP1, SCN1A, SCN2A and KCNQ2), (iv) enrichment of damaging URVs in NF1 gene in infantile spasms, and (v) comparison of DEE variation with published exomes from individuals with GGE and NAFE revealed three genes – SCN1A, KCNQ2 and ATP1A3 – commonly mutated in different types of epilepsy. This is an important contribution to the understanding of the genetic complexity of a severe neurological disorder for which genetic factors are now detectable in up to 40% of cases in some studies. The implication of second-hit contributions from non-DEE genes is a novel finding providing the first evidence that DEE is more genetically complex than previously thought and not an aggregate of simple Mendelian disorders, a concept that fits well with the established phenotypic heterogeneity. I suggest one major and several minor revisions.

Major Revision

The data analysis relies on comparison of URVs found in DEE cases and healthy individuals from the Japanese population with variation in publicly available databases. This is a large study presumably completed over many years and potentially prior to the advent of the gnomAD database (<http://gnomad.broadinstitute.org/>). However, comparison with this much larger database of healthy individuals from different populations should be completed. It contains 125,748 exome and 15,708 genome datasets, more than double the number compared to 60,706 exomes in ExAC, 6,503 exomes in the ESP exome variant server, and 3,554 genomes in ToMMo that have been used in this study. In addition, gnomAD includes a subset of exomes and genomes from East Asian populations including Japan. The URVs comparisons described in the manuscript should be repeated using both the entire gnomAD dataset, and a separate comparison with the subset of the gnomAD dataset derived from Japanese individuals, to ensure the five key findings summarized above are robust, and to better control for variant allele frequency differences between the normal Japanese population and other populations. These new comparisons are essential, and feasible because the data is publicly available. The use of gnomAD has become widespread since the release of the first dataset on August 18th 2016 (Lek et al 2016 Nature 536:285-291).

Minor Revisions

1. Variants observed once – page 6: the definition of URVs in this study could be interpreted as overly conservative or too strict – only observed once in the DEE cohort and never in public databases. There are many variants that occur only once in public databases, and presumably also in the author’s Japanese cohort reported here, that may or may not be sequence artefacts. How did

the authors handle this issue and how much Sanger validation of pathogenic or potentially pathogenic variants was completed in this study? Have potentially pathogenic (e.g. de novo) URVs been missed because the criteria are too strict? This is an issue already highlighted for epilepsy genes – in particular SCN1B – for which two proven pathogenic variants associated with febrile seizures and generalized epilepsy with febrile seizures plus are present more than once (n = 2 times) in ExAC (please refer to Bennett et al 2017 *Neurol Genet* 3(4):e163). This is an important issue that should be addressed in the manuscript including specifying the amount of URVs that were Sanger validated.

2. Moderate URVs – page 6: the use of this term is confusing and needs to be better defined. Does this mean missense or in frame variants that are predicted deleterious by in silico tools? Or are these variants distinct from other non-deleterious variants according to different criteria? The term ‘moderate’ or ‘moderate effect’ does not seem to imply that it is necessarily deleterious to protein function and therefore pathogenic or likely pathogenic. This needs to be clarified in the manuscript.

3. Gene burden analysis – page 15: the enrichment for NF1 variants in infantile spasms is striking. Was there enrichment for other non-DEE genes (e.g. ARGEF1, STXBP5L, HUWE1, CACNA1E) in specific DEE subtypes? This was not mentioned. It should also be noted that CACNA1E is now a reported DEE gene – please see very recent paper (Helbig et al 2018 *Am J Hum Genet* 103(5):666-678) that should be cited and the manuscript updated to reflect this. Mention in the Discussion on page 19 is not sufficient – CACNA1E should now be included as a DEE gene in the analyses. For the DEE patients with infantile spasms and confirmed de novo NF1 variants is brain imaging available? It would be helpful to include detailed phenotypic data for these three patients. Some phenotypic information is provided in the discussion on pages 18 and 19, but more comprehensive phenotypic details would be very helpful to the reader and should be included.

4. DEE subsets – page 20: it would be beneficial to briefly outline the DEE subsets most over- and under-represented in the study in the Discussion. It is clear that this is a significant limitation of this and similar studies.

Point-by-point responses to the Reviewers' comments

We first would like to sincerely thank all Reviewers for taking his/her time and effort. Our point-by-point responses to each comment can be found below. In this response letter, reviewer's comments are in *blue italic*, our replies are in **black**, and references from the revised manuscript are in *red italic*. In the updated files (Main Text, Supplementary Information and Supplementary Spreadsheets), the revised parts are shown in **red**.

Reviewer #1 (Remarks to the Author):

Takata and colleagues describe comprehensive exome-wide statistical investigations on 743 DEE cases compared to 2366 controls as well as to public databases. The approach and the calculations appear to be sound and extensive.

We thank this reviewer for appreciating soundness and comprehensiveness of our analyses.

However, I see some issues in the study design that need clarification or correction.

1) *The authors describe 743 cases with DEE. In the supplement, these cases are sub-divided into Infantile spasm, Ohtahara, Lennox Gastaut syndrome etc. of each a few hundred or dozens of individuals. Almost all these disease entities exhibit classic Epileptic Encephalopathies (EE). According to the most recent nomenclature of the ILAE, the term "Developmental and Epileptic Encephalopathies" (DEE) is used for disorders where developmental aspects occur prior to seizure onset. DEE is not meant to describe the whole group of EE and developmental disorders with epilepsy. Thus, "DEE" is a term on the same level as "EE" and "DEE" does not include "EE". I personally find this terminology unfortunate; however, the terms EE and DEE should be used accordingly throughout the manuscript (or be exchanged by another term).*

We appreciate this reviewer for pointing out this exceedingly important issue. We agree that subsets of specific epileptic syndromes in our cohort (e.g. infantile spasm, Ohtahara, Lennox Gastaut syndrome etc.) should be described as "EE", whereas "DEE" could be the best term to describe some of the individuals in the unspecified DEE group. Considering these, we decided to use "EE/DEE" in the revised manuscript instead of "DEE" (note that we use "DEE" in this response letter to avoid confusion of the other reviewers).

All 743 cases are of Japanese origin and are compared to 2,366 controls of Japanese origin as well as to ExAC (60,706 individuals), ESP (6,500 individuals) and ToMMo (1,070 individuals).

Before answering the specific points below raised by this reviewer, we would like to clarify that there would be a misunderstanding. In our study, we **did not** directly compare the 743 DEE cases with control databases. In our analyses, we used the data of ExAC, ESP and ToMMo just for filtering, by referring procedures that were demonstrated to be highly efficient in enriching neurodevelopmental disease-associated variants in previous studies (Genovese et al. Nature Neuroscience 2016, Kosmicki et al. Nature Genetics 2017 and Ganna et al. AJHG 2018). We also note that these filtering procedures are apparently very stringent, but were shown to be highly efficient as described above, whereas less-rare variants (e.g. doubleton/triplet variants or variants observed with a very low frequency in ExAC) were much "noisier" than singleton URVs. Indeed, among less-rare variants, expectable enrichment of damaging variants in disease groups was not clearly demonstrated in our and published studies (Genovese et al. Nature Neuroscience 2016 and Kosmicki et al. Nature Genetics 2017), suggesting that impact of disease-contributing variants can be masked by benign variants or low-frequency recurrent sequencing artifacts.

We used the data of general populations just for filtering because of the following reasons: i) it is totally improper to directly compare datasets from different ethnicities in statistical genetic analyses, and ii) it is also unacceptable to compare two datasets from different analytical/experimental pipelines (e.g. whole genome vs. whole exome) without extensive quality controls requiring individual-level data (such as PCA), even when both datasets are derived from the same ethnic group. As we have no way to access the individual-level data of general populations (including ToMMo and East Asians in ExAC, ESP and gnomAD), we did not perform a direct comparison of our cases with general East Asian populations in this study.

2) *Please provide more information on the 2,366 Japanese controls. Are they healthy individuals? Any other non-neurologic disorders? Relations?*

We are sorry for the insufficient information. Control individuals are, in a general sense, healthy individuals. However, they did not receive an in-detail assessment of non-neurologic disorders (e.g. metabolic syndrome, history of cancer etc.). We added this information as follows (**Page 29, Line 745**):

The vast majority of these individuals also have no severe non-neurologic disease at the time of recruitment, while they did not receive an in-detail assessment of non-neurologic disorders (e.g. metabolic syndrome, history of cancer etc.) for this study.

As it was described in **Method**, we confirmed that there are no apparently related individuals in our overall case-control cohort (**Page 31, Line 820** in the revised manuscript):

Regarding relationship among the studied 3,109 subjects, no pair was with PI_HAT (proportion of identity by descent, calculated by PLINK) > 0.2 , and 0.0001% (5/4,831,386) and 0.058% (2,812/4,831,386) of the pairs were with $PI_HAT > 0.15$ and > 0.1 , respectively.

3) *ExAC has been superseded by gnomAD (141,456 individuals, including all ExAC individuals with high sequencing quality). Thus, cases should be compared to gnomAD (with more individuals enabling better statistical power) and not ExAC.*

4) *ESP is a rather old database of minor quality compared to gnomAD. ESP could thus be omitted from the study.*

As we have clarified above, we did not directly compare our DEE cases with general populations in ExAC or ESP.

Nevertheless, we agree that use of the larger gnomAD data instead of ExAC provides more meaningful information, regardless of the purpose of the use of public databases. Also, a comparison between analyses with and without the gnomAD-based filtering should provide informative results. We therefore re-analyzed our whole dataset using the gnomAD data. We also note that we included *CACNA1E* in the list of known DEE genes in our re-analysis using the gnomAD data, as recommended by the Reviewer #3 (so, there were “58+1DEE genes” in the list for the re-analyses). The results are described in a new section as follows (**Page 19, Line 502**):

Confirmation of key findings by updating ExAC to gnomAD

Lastly, we performed confirmatory analyses by updating ExAC to the Genome Aggregation Database (gnomAD), focusing on the key findings in our study. By applying a further filtering using the gnomAD data (“non-neuro” subset of version 2.1, 104,068 exomes and 10,636 genomes), 29,841 out of the 211,988 URVs

(14.1%) in 3,109 individuals were removed. In these confirmatory analyses, we also included CACNA1E in the list of established DEE genes (so, there were “58+1DEE genes” in the list for the confirmatory analyses).

When we repeated the overall enrichment analyses of various functional types of URVs in DEE (in **Figs. 1 and 2**, 66 tests in total), we largely replicated the findings in analyses without the gnomAD-based filtering (**Supplementary Fig. 9**, correlation coefficient > 0.99 for both $-\log_{10}$ P-values and \log_2 ORs in the 66 tests in **Supplementary Table 7 or 16**).

We found that the additional gnomAD-based filtering led to an increase of OR in the majority of the tests (43/66, significantly larger than 0.5 at $P = 0.019$ when each test was treated as independent, two-tailed binomial test), indicating that the additional filtering has contributed to further concentration of disease-associated URVs. An increase of OR was clear in URV categories expected to be enriched for primary pathogenic variants, such as LOF or missense URVs in 58+1DEE genes. On the other hand, a decrease of ORs was observed in categories likely harboring URVs contributing to DEE in an oligogenic manner (e.g. non-58+1DEE gene dURVs in the cases with pURVs in 58+1DEE genes), may suggesting that further stringent filtering causes loss of these modifier URVs (**Supplementary Table 16**).

*was little change in the analysis applying the additional gnomAD-based filtering (see “Change_in_P_vs_non-pURV_carriers” column of **Supplementary Table 17**). Regarding NF1, we found that all three damaging DNMs (c.3445A>G [p.Met1149Val], c.4835+1G>T and c.5330T>A [p.Val1777Asp]) were not found in the gnomAD non-neuro dataset. Therefore, exome-wide significant enrichment of damaging NF1 DNMs in infantile spasm was unchanged. We also confirmed significant or nominally significant enrichment of dURVs in genes commonly mutated in different types of epilepsies (SCN1A, KCNQ2, ATP1A3 and GRIA4, **Supplementary Table 17**).*

Therefore, i) findings in analyses without the gnomAD-based filtering were in general confirmed in the analyses using the additional gnomAD-based filtering (**Supplementary Fig. 9** and **Supplementary Table 16**), ii) the additional gnomAD-based filtering (unfortunately) did not contribute to identification of genes newly reached to the exome-wide significance threshold in the gene-based burden test (**Supplementary Table 17**), and iii) application of an additional filtering likely contributes to further concentration of primary disease-causing URVs, while this may cause loss of some oligogenic/modifier URVs.

Regarding ESP, we agree that ESP is a rather old database. However, we believe that use of an additional large-scale dataset provides something when the data is used just for filtering. Therefore, we could not find any good reason to omit this database. Of course, we agree that an old dataset generated by an old technology and an analytical pipeline should not be directly compared with a newer dataset (without extensive quality controls and normalization).

5) The most important control cohort appears to be the (not further described) series of 2,366 as well as ToMMo as both comprise exclusively Japanese controls. Comparison of 743 cases versus 2,366 + 1,070 Japanese controls does not allow for robust statistical calculations.

Again, we note that we used the ToMMo data just for filtering and did not perform an analysis of 743 cases versus 2,366 + 1,070 Japanese controls. It is impossible to directly compare our 743 cases with the Japanese individuals of ToMMo, because i) our data were generated by exome sequencing, while ToMMo provides variant data from whole genome sequencing, ii) ToMMo provides a list of high confidence variants that passed their own stringent filtering criteria that is different from the GATK’s best practice, iii) ToMMo only provides summary data for each variant (e.g. minor allele frequency etc.), and iv) the vast majority of the individuals included in ToMMo were recruited from Tohoku region of Japan, whereas our cases and controls were from various areas of Japan (thus, quality controls using individual-level data are mandatory).

Regarding the robustness of statistical calculations, we would like to clearly claim that this should be evaluated with the level of significance and effect size (and not with an impression on the sample size). As we have clearly mentioned, findings in **Figures 1 and 2** that remain significant after performing multiple testing correction (shown in **Supplementary Table 7**), and genes with exome-wide significance in **Figure 3** should be considered as statistically robust. We agree that other findings with marginal robustness (e.g. enrichment of dURVs in nominal genes) are inconclusive with the current sample size and should be evaluated in independent or expanded cohorts. We also understand that replication/confirmatory studies are always important regardless of the statistical significance and effect size in the initial study, as we have clarified as follows (**Page 25, Line 658**):

Nevertheless, replication studies in independent cohorts are warranted to further evaluate validity of this finding, and to test if other results that only showed nominal significance in our current analyses are true or not.

Therefore, despite ExAC or gnomAD, all ultra-rare variants found in 743 cases can very well be simply rare Japanese variants and not be disease-associated at all. Even including gnomAD would only marginally solve this problem as it includes only 152 Japanese alleles. How do the authors make sure that rare variants are not simply population-specific?

First, we never state that all URVs in DEE are disease-associated. We rather claimed that some of (not all of) the specific types of URVs (e.g. LOF/null URVs) are actually associated with the risk of DEE. For instance, in the case of LOF (null) URVs in all genes, we observed that these URVs were ~1.1 times more frequent in DEE when compared with controls (**Figure 1a**). This means that at least ~10% of LOF URVs in DEE are expected to contribute to the disease risk (see below), and the null hypothesis was rejected at a high-significance level (P = 0.000017). Therefore, we can assure that some of the LOF URVs are actually contributing to the risk of DEE.

Regarding the proportion of LOF URVs contributing to the risk of DEE, our result can be explained by one of the following scenarios: i) all LOF URVs excessive in DEE are completely penetrant, and the proportion of LOF URVs in DEE conferring the disease risk is ~10%, ii) each LOF URVs increases the risk of DEE 1.1 times, and the proportion of LOF URVs in DEE conferring the disease risk is 100%, and iii) LOF URVs in DEE is a mix of variants with various effect sizes (e.g. completely penetrant, moderately penetrant, with a weak effect, or with no effect), and the proportion of LOF URVs in DEE conferring the disease risk is a value within the range of 10-100%. Therefore, in any case, we are sure that at least ~10% of LOF URVs in DEE contribute to the disease risk. On the other hand, given existence of highly penetrant LOF URVs (e.g. those in known DEE genes), the scenario #ii is unrealistic. So, we never state that all URVs in DEE are disease-associated.

6) *To evaluate the previous question – how many ultra-rare variants are found in the Japanese controls that are not reported in public databases?*

We observed a total of 127,766 coding URVs not in databases among 2,366 controls (54.0 URVs on average). The distribution of per-individual URV counts is also visually displayed in **Supplementary Figure 1**.

7) *How significant would be the opposite calculation – comparing ultra-rare variants in databases versus Japanese cases?*

As we described above, we used the data in database for filtering. Therefore, no variant in databases was considered as URV in this study (regardless of the case-control status). The number of coding URVs in 743 cases was 41,248 (55.5 URVs on average). The distribution of per-individual URV counts is visually displayed in **Supplementary Figure 1**.

8) *Multiple testing for 66 hypotheses appears to be low and should be considerably increased.*

The number of hypotheses tested in **Figures 1 and 2** is exactly 66. Hence, we are confident that this is definitely the correct number. Note that we did not perform exome-wide gene-based analyses, in which a

much larger number of hypotheses are usually tested, in Figures 1 and 2. Rather, we performed analyses of “gene sets (i.e. the sets of high-pLI genes, 58DEE genes, or non-58DEE genes)”. We further clarified this as follows (**Page 9, Line 237**):

We next analyzed the set of the other non-58EE/DEE genes...

If this reviewer (still) thinks that the number (66) is incorrect, we would like to ask him/her about a number that would be more appropriate.

9) *According to ACMG, “null variants” should be preferred over “loss-of-function variants”.*

We appreciate this suggestion. We substituted “LOF variants” with “null variants”. As the “probability of being LOF-intolerant (pLI) score” is a defined term by the developer, we use LOF for description of this scoring system. Also please note that in this response letter we use LOF to prevent confusion.

10) *“Second-hit” effects is possibly not, what the authors intend to say. I believe “oligogenic” effects would be more suitable, is that right?*

We appreciate this excellent advice. We agree that “oligogenic effect” is the more appropriate word. We use “oligogenic” in the revised manuscript instead of “second-hit”. In some parts, we also use the term “modifier”.

Reviewer #2 (Remarks to the Author):

In the study by Takata et. al. they perform Ultra Rare Variant (URV) case control analysis in a cohort of 743 Developmental and epileptic encephalopathy (DEE) cases. They found a significant excess of damaging URV within known DEE genes, which is not surprising. They also observed an excess of damaging URV in non-DEE genes. Gene-based burden analysis was performed with five known DEE genes being exome-wide significant. The NF1 gene which was nominally significant was a novel finding and 3 variants were validated as de-novo and upon re-analysis was found to be significant.

We appreciate this reviewer for succinctly summarizing key findings of our study.

General comment

This manuscript was very difficult to read as it was verbose and was mixed with expected results and potentially interesting results. I found that the expected and therefore uninteresting results took a disproportionate amount of space, leaving the reader frustrated that they have to search very hard for the interesting results. For example, the discovery of excess of damaging variants in known DEE genes is expected. The gene-based burden test showing five known DEE genes is an expected result.

We thank this valuable general comment. As you will see, we moved substantial parts to **Supplementary Information**. We also have shortened the **Abstract**.

Comments

1. The excess of damaging URV in non-DEE genes is an interesting and novel result. However, there is little explanation in the results and discussion on what genes make up this gene list. It is buried in the methods and currently is difficult to interpret how it was generated. Also, how were the significance thresholds determined for tests involving non-DEE genes?

We are sorry for lack of explanation in our manuscript. Non-58DEE genes are all the genes except for the 58 DEE genes. Specifically, there were 18,338 non-DEE protein-coding genes (based on ENSG IDs annotated by SnpEff) with one or more URVs in the overall cohort. We clarified this as follows (**Page 9, Line 237**):

Before

On the other hand, when we analyzed URVs in the other non-58DEE genes, still there was...

After

We next analyzed the set of the other non-58DEE genes. In our overall cohort, 18,396 protein-coding genes (based on ENSG IDs annotated by SnpEff) were with one or more URVs, and thereby there were 18,338 (18,396-58) analyzable non-58DEE genes. We found that...

Regarding the significance threshold, we uniformly applied multiple testing correction to each of the URV types analyzed in Figures 1 and 2. As there were a total 66 hypotheses tested in **Figures 1 and 2**, we used this number. Therefore, the significance threshold for the analyses in Figures 1 and 2 was Bonferroni-corrected or Benjamini-Hochberg-corrected $P < 0.05$, regardless of whether we test URVs in 58DEE genes or non-58DEE genes. Detailed results are shown in **Supplementary Table 7**. The corresponding raw P value was 0.000758 for Bonferroni correction and ~ 0.02 for Benjamini-Hochberg correction. These raw P value thresholds were clarified as described in our response to the comment #6 by this reviewer. Also please note that we did not perform analyses of each non-DEE gene in Figures 1 and 2; we rather analyzed a single “gene set” of non-DEE genes.

In the tests of individual genes in **Figure 3**, we uniformly applied the exome-wide significance

threshold considering that there are ~20,000 protein-coding genes (regardless of whether a gene is 58DEE gene or not).

2. If there is nothing interesting, the analysis of doubletons and tripletons should be moved to the supplementary and condensed to a 1-2 lines in the main text instead of being 1.5 pages.

We moved this section to **Supplementary Information**. In **Main Text**, we briefly summarized the results as follows (**Page 12, Line 333**):

*We also analyzed doubleton and tripleton rare variants, that is, variants observed twice and three times in our overall case-control cohort, respectively, and not seen in any of ExAC, ESP, and ToMMo databases. As a whole, we did not observe overall enrichment of any functional types of doubleton/triplet variants in DEE (**Supplementary Fig. 5**). The total number of convincingly pathogenic doubleton/triplet alleles ($n = 24$, **Supplementary Table 9**) was smaller than singleton pURVs ($n = 116$). More detailed results of analyses of doubleton/triplet rare variants are described in **Supplementary Text**.*

3. Similarly, exome-wide association of SNPs should be moved to the supplementary as there are no interesting results and shouldn't consume 0.5 pages.

We moved this section to **Supplementary Information**. In **Main Text**, we briefly summarized the results as follows (**Page 14, Line 382**):

*When we performed an exome-wide association study of single nucleotide polymorphisms (SNPs: defined as variants with a minor allele frequency [MAF] > 1%), we found that there is no exonic SNP surpassing the genome-wide significance threshold ($P < 5 \times 10^{-8}$) (**Supplementary Fig. 6 and Supplementary Table 10**). More detailed results of the exome-wide association study, including calculation of statistical power (**Supplementary Fig. 7**) is shown in **Supplementary Text**.*

4. Lastly, the same applies for "Comparison between DEE and common epilepsies". The results of the analysis is nominally significant and the novel result of de novo variants in NF1 would be more interesting than ending with another uninteresting result.

We moved this section to **Supplementary Information**. In **Main Text**, we put the brief summary of these analyses before the section for *NF1* mutations as follows (**Page 16, Line 436**):

*By comparing the results of gene-based burden tests for DEE in our study and a recent study for common forms of epilepsy (genetic generalized epilepsy: GGE and non-acquired focal epilepsy: NAFE), we found *SCN1A*, *KCNQ2*, *ATP1A3* and *GRIA4* as genes with nominally significant burden of rare damaging variants in both DEE and common epilepsy (**Supplementary Fig. 8a and Supplementary Table 13**). We also found significant gene-level overlap of DEE with GEE (**Supplementary Fig. 8b**, $P = 0.0115$), but not with NAFE (**Supplementary Fig. 8b**, $P = 0.373$) (see **Supplementary Text** for more details).*

5. The above should reduce the large size of the abstract that's littered with uninteresting results and unimportant conclusions.

We greatly appreciate this reviewer's above comments that make our manuscript much more easy-to-read.

5. In the first paragraph of the discussion, the comments of doing a separate analysis instead of contributing to a better powered consortium analysis I don't agree with entirely. However, I would like the authors to expand on the last comment, "we conducted more various statistical analyses to better understand the genetic architecture of DEE" and be explicit on the exact analysis and why this would not have been possible in the Zhu et al. study?

We explicitly described some specific analyses performed in our study besides the gene-based burden test as follows (**Page 21, Line 563**, the changed part is underlined):

while in our study we conducted more various statistical analyses besides gene-based burden test, such as enrichment analysis of various types of URVs in DEE individuals with or without pathogenic mutations, GO enrichment analysis and others, to better understand the genetic architecture of DEE.

These analyses besides the gene-based burden test might be able to be done with the data in Zhu et al., when appropriate normalization/quality controls are performed. However, it is unable to determine if it is feasible or not without checking their raw data. Therefore, we can say nothing about whether it is possible or not as of this moment. We also note that we are discussing about participation to a larger international consortium after publication of this and some other reports.

6. Apart from exome-wide significance, throughout the manuscript it is difficult to know what the threshold for significance after adjusting for multiple test correction. It is not explicitly stated in the results text or indicated on the figures. Furthermore, there should be an indication of significance on the figures.

We are sorry for this unclearness. To address this, we first added the thresholds for raw P value when Bonferroni or Benjamini-Hochberg procedure is applied in **Main Text (Page 11, Lines 297 and 305)**. Please note that these significance thresholds can be determined only after the completion of the analyses in **Figures 1 and 2**. Second, we added these thresholds in the legends of **Figures 1 and 2**. Third, we indicated enrichment with Bonferroni-corrected $P < 0.05$ and Benjamini-Hochberg-corrected $P < 0.05$ by double and single asterisks, respectively, in **Figures 1 and 2** (an example is shown below).

7. A comment or discussion should be made on the odds ratio and how this compares with previous results. This makes up the bulk of the other dimension (i.e. x-axis) of the data presented in Figures 1 and 2.

First, we would like to note that we are not confident whether we correctly understand the intention of this

comment. Specifically, we are uncertain what the term “previous results” refers to. If “previous results” indicate the results in the Zhu et al. study, they did not perform enrichment analysis of different types of variants (e.g. those shown in **Figures 1 and 2** of our manuscript) at all. We also added a clearer explanation on the interpretation of odds ratios as follows (**Page 39, Line 1,041**):

i.e. one additional URV changes the risk of being DEE with the indicated odds ratio

Lastly, we note that we added some new sections to answer comments by the other reviewers. We are willing to discuss with the reviewers and editors if these new sections should be also moved to **Supplementary Information**.

Reviewer #3 (Remarks to the Author):

Takata et al report a comprehensive genetic analysis study designed to understand the genetic architecture of developmental and epileptic encephalopathy (DEE). This is a very large cohort study of a severe disabling disorder from a single population. An impressive cohort of 743 DEE cases and 2,366 healthy individuals from Japan received exome analysis. The analysis strategy focused on ultra-rare variants (URVs) observed only once in this cohort and not seen in public databases. The key findings were: (i) damaging (LOF or deleterious missense) URVs were significantly enriched in DEE cases, in both known DEE and non-DEE genes, (ii) enrichment in non-DEE genes remained even after removal of a subset of cases with convincingly pathogenic variants suggesting “second hit” effects in non-DEE genes, (iii) significant burden of damaging URVs in five known DEE genes (CDKL5, STXBP1, SCN1A, SCN2A and KCNQ2), (iv) enrichment of damaging URVs in NF1 gene in infantile spasms, and (v) comparison of DEE variation with published exomes from individuals with GGE and NAFE revealed three genes – SCN1A, KCNQ2 and ATP1A3 – commonly mutated in different types of epilepsy. This is an important contribution to the understanding of the genetic complexity of a severe neurological disorder for which genetic factors are now detectable in up to 40% of cases in some studies. The implication of second-hit contributions from non-DEE genes is a novel finding providing the first evidence that DEE is more genetically complex than previously thought and not an aggregate of simple Mendelian disorders, a concept that fits well with the established phenotypic heterogeneity. I suggest one major and several minor revisions.

We thank this reviewer for his/her careful reading of our manuscript and appreciation of the novelty of our finding.

Major Revision

The data analysis relies on comparison of URVs found in DEE cases and healthy individuals from the Japanese population with variation in publicly available databases. This is a large study presumably completed over many years and potentially prior to the advent of the gnomAD database (<http://gnomad.broadinstitute.org/>). However, comparison with this much larger database of healthy individuals from different populations should be completed. It contains 125,748 exome and 15,708 genome datasets, more than double the number compared to 60,706 exomes in ExAC, 6,503 exomes in the ESP exome variant server, and 3,554 genomes in ToMMo that have been used in this study. In addition, gnomAD includes a subset of exomes and genomes from East Asian populations including Japan. The URVs comparisons described in the manuscript should be repeated using both the entire gnomAD dataset, and a separate comparison with the subset of the gnomAD dataset derived from Japanese individuals, to ensure the five key findings summarized above are robust, and to better control for variant allele frequency differences between the normal Japanese population and other populations. These new comparisons are essential, and feasible because the data is publicly available. The use of gnomAD has become widespread since the release of the first dataset on August 18th 2016 (Lek et al 2016 Nature 536:285-291).

We appreciate this reviewer’s comment on a very important point. Before answering this comment, we would like to clarify that in our analyses we just used the data of general populations (i.e. ExAC, ESP and ToMMo) for “filtering”. In our study, we did not perform a direct comparison between our cases and general populations in these databases. To perform accurate statistical assessment, it is totally improper to compare datasets from different ethnicities. In addition, it is also unacceptable to compare two datasets from different analytical/experimental pipelines (e.g. whole genome vs. whole exome) without extensive quality controls requiring individual-level data (such as PCA) even when both datasets are derived from the same ethnic group.

Nevertheless, it is true that the ExAC can be updated by the larger gnomAD data, and it is highly important to test if the results retain significant when the newer gnomAD data are used (regardless of the purpose of the use of the data of general populations). Also, a comparison between analyses with and without gnomAD-based filtering should provide informative results. We therefore re-analyzed our whole dataset using the gnomAD data. We also included *CACNA1E* in the list of known DEE genes in our re-analysis using the gnomAD data (so, there were “58+1DEE genes” in the list for the re-analyses). The results are described in a new section as follows (Page 19, Line 502):

Confirmation of key findings by updating ExAC to gnomAD

Lastly, we performed confirmatory analyses by updating ExAC to the Genome Aggregation Database (gnomAD), focusing on the key findings in our study. By applying a further filtering using the gnomAD data (“non-neuro” subset of version 2.1, 104,068 exomes and 10,636 genomes), 29,841 out of the 211,988 URVs (14.1%) in 3,109 individuals were removed. In these confirmatory analyses, we also included *CACNA1E* in the list of established DEE genes (so, there were “58+1DEE genes” in the list for the confirmatory analyses).

When we repeated the overall enrichment analyses of various functional types of URVs in DEE (in Figs. 1 and 2, 66 tests in total), we largely replicated the findings in analyses without the gnomAD-based filtering (Supplementary Fig. 9, correlation coefficient > 0.99 for both $-\log_{10}$ P-values and \log_2 ORs in the 66 tests in Supplementary Table 7 or 16).

Supplementary Figure 9. Plots of results in enrichment analyses of various types of URVs in DEE with or without gnomAD-based filtering.

(a) The $-\log_{10}$ P-values obtained in analyses without gnomAD-based filtering (X-axis, Supplementary Table 7) and those obtained in analyses with gnomAD-based filtering (Y-axis, Supplementary Table 16) were plotted. (b) The \log_2 ORs obtained in analyses without gnomAD-based filtering (X-axis, Supplementary Table 7) and those obtained in analyses with gnomAD-based filtering (Y-axis, Supplementary Table 16) were plotted. The red dots indicate the analyses for which a smaller (more significant) P-value or a larger OR was observed when gnomAD-based filtering was applied. The blue dots indicate the analyses for which a larger (less significant) P-value or a smaller OR was observed with gnomAD-based filtering. The gray dotted line indicates $X = Y$. Correlation coefficient (r) of X and Y is shown in each panel.

Among the specific tests, we confirmed enrichment of dURVs in DEE cases, in both 58+1DEE genes (LOF; Bonferroni-corrected $P = 1.03 \times 10^{-16}$ and CD missense; Bonferroni-corrected $P = 6.38 \times 10^{-25}$) and the other non-58+1DEE genes (dURV; Bonferroni-corrected $P = 0.00139$). Enrichment of non-58+1DEE gene dURVs in the subset of 118 DEE cases with pURVs in 58+1DEE genes showed an OR similar to that observed before the gnomAD-based filtering (1.119 and 1.125 in the analyses with and without the filtering, respectively) with a P-value (raw- $P = 0.000922$) that remained significant after Benjamini-Hochberg correction (corrected- $P = 0.00468$) but not after Bonferroni adjustment (corrected- $P = 0.0608$), possibly due to loss of potential oligogenic URVs as mentioned above.

*By repeating the gene-based burden test (in **Fig. 3** and **Supplementary Fig. 8**) with the gnomAD-based filtering, we confirmed exome-wide significant enrichment of dURVs in five known DEE genes (CDKL5, STXBP1, SCN1A, SCN2A and KCNQ2), while there was no gene newly reached to the significance threshold by application of the additional filtering (**Supplementary Table 17**). In general, there was little change in the analysis applying the additional gnomAD-based filtering (see “Change_in_P_vs_non-pURV_carriers” column of **Supplementary Table 17**). Regarding NF1, we found that all three damaging DNMs (c.3445A>G [p.Met1149Val], c.4835+1G>T and c.5330T>A [p.Val1777Asp]) were not found in the gnomAD non-neuro dataset. Therefore, exome-wide significant enrichment of damaging NF1 DNMs in infantile spasm was unchanged. We also confirmed significant or nominally significant enrichment of dURVs in genes commonly mutated in different types of epilepsies (SCN1A, KCNQ2, ATP1A3 and GRIA4, **Supplementary Table 17**).*

Briefly summarizing the above results focusing on the five key findings of this study, the following four key findings, *(i) damaging (LOF or deleterious missense) URVs were significantly enriched in DEE cases, in both known DEE and non-DEE genes, (iii) significant burden of damaging URVs in five known DEE genes (CDKL5, STXBP1, SCN1A, SCN2A and KCNQ2, (iv) enrichment of damaging URVs in NF1 gene in infantile spasms, and (v) comparison of DEE variation with published exomes from individuals with GGE and NAFE revealed three genes – SCN1A, KCNQ2 and ATP1A3 – commonly mutated in different types of epilepsy, were perfectly confirmed.*

Regarding the key finding *(ii) enrichment in non-DEE genes remained even after removal of a subset of cases with convincingly pathogenic variants suggesting “second hit” effects in non-DEE genes*, this was confirmed at the level of Benjamini-Hochberg-corrected significance threshold, but not at the level of Bonferroni-corrected threshold.

Given that Bonferroni procedure is too stringent considering extensive dependence among each of the hypotheses tested in **Figures 1 and 2**, and that similar effect sizes (i.e. OR) were observed in the analyses with and without the gnomAD-based filtering, the result should be, in a general sense, considered as still significant. Meanwhile, we understand necessity of replication studies, as stated below (**Page 25, Line 658**):

Nevertheless, replication studies in independent cohorts are warranted to further evaluate validity of this finding, and to test if other results that only showed nominal significance in our current analyses are true or not.

Methods section was updated as follows (Page 37, Line 1,025):

Confirmatory analyses using the gnomAD data

We repeated the enrichment analyses of various functional types of URVs in DEE (in Figs. 1 and 2) and the gene-based burden test (in Fig. 3 and Supplementary Fig. 8) using the data of URVs to which a further filtering based on the gnomAD dataset was applied. In this analysis, we used the “non-neuro” dataset of gnomAD, which include only samples from individuals who were not ascertained for having a neurological condition in a neurological case/control study (104,068 exomes and 10,636 genomes in total). By applying this additional filtering, 24,246/169,014 (14.3%) coding and 5,595/42,974 (13.0%) non-coding URVs were removed.

Minor Revisions

1. Variants observed once – page 6: the definition of URVs in this study could be interpreted as overly conservative or too strict – only observed once in the DEE cohort and never in public databases. There are many variants that occur only once in public databases, and presumably also in the author’s Japanese cohort reported here, that may or may not be sequence artefacts. How did the authors handle this issue and how much Sanger validation of pathogenic or potentially pathogenic variants was completed in this study? Have potentially pathogenic (e.g. de novo) URVs been missed because the criteria are too strict? This is an issue already highlighted for epilepsy genes – in particularly SCN1B – for which two proven pathogenic variants associated with febrile seizures and generalized epilepsy with febrile seizures plus are present more than once (n = 2 times) in ExAC (please refer to Bennett et al 2017 Neurol Genet 3(4):e163). This is an important issue that should be addressed in the manuscript including specifying the amount of URVs that were Sanger validated.

It is possible that some potentially pathogenic URVs have been missed because of our strict criteria. However, as we have stated in our manuscript, the main purpose of this study is to obtain a deeper insight into the overall genetic landscape of DEE by utilizing various statistical approaches. In this context, we applied the strict criteria, whose effectiveness were demonstrated in previous studies (Genovese et al. Nature Neuroscience 2016, Kosmicki et al. Nature Genetics 2017 and Ganna et al. AJHG 2018), in order to exclude likely benign variants and sequence artifacts as far as possible. Indeed, less-rare variants (e.g. doubleton/tripleton variants or variants observed with a very low frequency in ExAC) in our and published studies were much “noisier” than singleton URVs, and thereby expectable enrichment of damaging variants in disease groups was not clearly demonstrated in our and published studies (Genovese et al. Nature Neuroscience 2016 and Kosmicki et al. Nature Genetics 2017), suggesting that impact of disease-contributing less-rare variants can be masked by benign variants or low-frequency recurrent sequencing artifacts. We therefore would like to strongly claim that use of very strict criteria is the first choice for accurate statistical assessment of overall disease genetic architecture (e.g. analyses addressing which “type” of rare variants contribute to a disease).

Meanwhile, it is true that some genuine DEE-contributing variants observed in databases of general populations at a low frequency would be missing from the list of URVs in our study. We clarified this point by citing the paper kindly indicated by this reviewer as follows (Page 6, Line 170, the changed part is underlined).

While we were aware that recurrent pathogenic mutations and true disease-contributing variants observed in general populations at a low frequency (Bennett et al 2017) would be removed by restricting the primary

scope of our analysis to these URVs, this procedure enables us to do an unbiased analysis that can efficiently detect enrichment of rare damaging variants in other neurodevelopmental disorders

Regarding the amount the URVs subjected to Sanger validation, we confirmed the following URVs by Sanger sequencing: i) all pURVs in 58DEE genes (n = 116, **Supplementary Table 5**), ii) all *de novo* dURVs in *NF1*, and iii) all *de novo* or hemizygous dURVs in the selected candidate genes (*ARFGEF1*, *STXBPL5L*, *HUWE1* and *CACNA1E*). The other URVs were not analyzed by Sanger sequencing. All of these URVs were Sanger-confirmed (thus, the validation rate = 100%). This validation rate ensures high accuracy of our variant call, and would also indicates that virtually all of the damaging URV calls observed in known or strong candidate DEE genes in DEE patients are inevitably true positives. The contents of the Sanger-validated URVs were described in **Methods**. We also clarified that dURVs in non-58DEE genes that were not considered as pURV (138 - 116 =22 variants) were not excluded due to a failure in Sanger validation, but because of their inheritance from one of the unaffected parents, lack of DNA samples of the parent, or mosaicism (we conservatively did not include mosaic variants into the group of pURVs) (**Page 33, Line 877**).

2. Moderate URVs – page 6: the use of this term is confusing and needs to be better defined. Does this mean missense or in frame variants that are predicted deleterious by in silico tools? Or are these variants distinct from other non-deleterious variants according to different criteria? The term ‘moderate’ or ‘moderate effect’ does not seem to imply that it is necessarily deleterious to protein function and therefore pathogenic or likely pathogenic. This needs to be clarified in the manuscript.

We thank this reviewer for mentioning this important point that leads to confusion for the readers. As shown in **Supplementary Table 2** and **Methods**, and originally defined by SnpEff (“Effect prediction details” section of http://snpeff.sourceforge.net/SnpEff_manual.html), Moderate variants consist of disruptive inframe deletion, disruptive inframe insertion, inframe deletion, inframe insertion, missense variant, protein protein contact, and start lost variants. All missense/inframe variants were included regardless of their predicted deleteriousness by in silico tools. The term “disruptive” for inframe indels means that one codon is changed and one or many codons are inserted/deleted. “Protein protein contact” indicates variants at protein-protein interaction loci.

To further clarify the definition of Moderate variants, we i) stated that all missense/inframe variants were included in this type regardless of their predicted deleteriousness by in silico tools at this stage of classification, and ii) added the link to the SnpEff effect prediction details page as follows (**Page 6, Line 177**):

*we stratified these variants according to their functionality into loss-of-function (LOF: nonsense, frameshift, splice site and read-through), Moderate (defined by SnpEff; e.g. missense and inframe indel; all missense/inframe variants were included in this type regardless of their predicted deleteriousness by in silico tools at this stage of classification), synonymous and non-coding variants (see **Supplementary Table 2, Methods and SnpEff manual page [“Effect prediction details” section of http://snpeff.sourceforge.net/SnpEff_manual.html]** for details).*

3. Gene burden analysis – page 15: the enrichment for NF1 variants in infantile spasms is striking. Was there enrichment for other non-DEE genes (e.g. ARGEF1, STXBP5L, HUWE1, CACNA1E) in specific DEE subtypes? This was not mentioned.

It is true that DEE subtypes of individuals with a dURV in *ARGEF1*, *STXBP5L*, *HUWE1* or *CACNA1E* were only shown in **Supplementary Table 14** and not in **Main Text**. We added this information and some discussion as follows (**Page 23, Line 619**):

*Among the DEE cases with a dURV in the other genes subjected to an additional analysis (*ARGEF1*, *STXBP5L*, *HUWE1* and *CACNA1E*), we did not observe statistically significant enrichment of a specific DEE subset under the hypergeometric distribution. However, we note that two cases with a confirmed DNM in *CACNA1E* and another case with a previously reported *CACNA1E* dURV (c.1054G>A [p.Gly352Arg]; thus, this variant fulfill the criteria for “Likely Pathogenic” in the ACMG guideline) were all diagnosed with infantile spasm. Also, three cases with a hemizygous (two maternally inherited variants and one DNM) CD missense URV in *HUWE1* were all with infantile spasm. These observations could be at least suggestive of the phenotypic specificities in individuals carrying likely pathogenic variants in *CACNA1E* and *HUWE1*.*

*It should also be noted that *CACNA1E* is now a reported DEE gene – please see very recent paper (Helbig et al 2018 Am J Hum Genet 103(5):666-678) that should be cited and the manuscript updated to reflect this. Mention in the Discussion on page 19 is not sufficient – *CACNA1E* should now be included as a DEE gene in the analyses.*

We agree that it was insufficient to just cite the paper by Helbig et al. (ref. 43 in the previous version of the manuscript). To address this, i) we added more extensive discussion about this point as shown below (**Page 18, Line 470**), and ii) we briefly summarized phenotypes of two individuals with a confirmed *CACNA1E* DNM in **Supplementary Table 15** (note that the c.2104G>A [p.Ala702Thr] variant was newly confirmed as a DNM after the initial submission of this manuscript).

*we confirmed that two CD missense URVs in *CACNA1E*, a gene recently reported as a DEE/neurodevelopmental disorder gene, are DNMs. Of the confirmed two DNMs in this study, the c.2104G>A [p.Ala702Thr] variant was reported as recurrent mutations in the above mentioned study, and the c.2092T>C [p.Phe698Leu] variant was a mutation at an amino acid residue where another substitution (p.Phe698Ser) was reported. In accordance with the recently reported cases with *CACNA1E* mutations, we observed congenital contractures and dystonia in both of our cases (brief summary of the clinical phenotypes is available in **Supplementary Table 15**). Taken these together, pathogenicity of these damaging *CACNA1E* DNMs is quite convincing, and this gene should be considered as an established DEE gene.*

We also agree that now *CACNA1E* should be considered as an established DEE gene. As described above, we included *CACNA1E* in the list of known DEE genes in our re-analysis using the gnomAD data (so that, two DEE cases with a confirmed *CACNA1E* DNM were added to the group of individuals with pURV, and *CACNA1E* was removed from the list of non-58DEE genes).

For the DEE patients with infantile spasms and confirmed de novo NF1 variants is brain imaging available? It would be helpful to include detailed phenotypic data for these three patients. Some phenotypic information is provided in the discussion on pages 18 and 19, but more comprehensive phenotypic details would be very helpful to the reader and should

be included.

Thank you very much for this meaningful recommendation. We added brief overview of the clinical manifestations, including the existence/absence of brain tumors, in the three individuals with a damaging *NF1* DNM in **Supplementary Table 15**.

4. DEE subsets – page 20: it would be beneficial to briefly outline the DEE subsets most over- and under-represented in the study in the Discussion. It is clear that this is a significant limitation of this and similar studies.

We greatly appreciate this reviewer for pointing out this important limitation. We added information on DEE subsets over- or under-represented in our cohort as follows (**Page 25, Line 664**).

Another important point that should be considered is that specific subtypes of DEE would be over- or under-represented in our DEE cohort. This makes it difficult to compare the results of our and other studies. In particular, our cohort includes cases of Ohtahara or Doose syndromes, who had been recruited for the specific projects, and thereby is expected to be enriched for these subtypes. When our cohort was compared with other studies for the prevalence of DEE in Japan (Okayama district) or far-east Asia, proportions of unspecified DEE (37, 37 and 40 % in our study, ref. 51 and ref. 52, respectively) and infantile spasms (36, 39 and 36 %) were similar across studies. On the other hand, there were expectable over-representations of Ohtahara syndrome (12, 0 and 1%) and Doose syndrome (3, 1 and 1 %) in our cohort. Early myoclonic encephalopathy (3, 1 and 0 %) and migrating partial seizures of infancy (4, 0 and 0 %) were also more frequent in our cases, while Lennox-Gastaut syndrome (3, 5 and 11 %) was less frequent. This information should be taken into account in future meta/mega-analyses incorporating our data.

REVIEWERS' COMMENTS:

Reviewer #1 (Remarks to the Author):

The authors adequately addressed all issues rose by the reviewers and also clarified some misunderstandings. The manuscript has improved significantly. I have no additional comment.

Reviewer #2 (Remarks to the Author):

No additional Comments. Authors have fully answered and addressed my comments.

Reviewer #3 (Remarks to the Author):

The authors have carefully and comprehensively addressed all of my concerns. As a result their manuscript is improved.